# The effect of horseshoes and surfaces on horse and jockey centre of mass displacements at gallop

Kate Horan[1,2‡]*, Kieran Kourdache[3�], James Coburn[4�], Peter Day[1], Henry Carnall[4‡], Dan Harborne[4‡], Liam Brinkley[4‡], Lucy Hammond[3], Sean Millard[1], Bryony Lancaster[2], Thilo Pfau[1‡]

**1** The Royal Veterinary College, Hatfield, Hertfordshire, United Kingdom, **2** The Royal (Dick) School of Veterinary Studies, The University of Edinburgh, Easter Bush Campus, Midlothian, United Kingdom, **3** The British Racing School, Newmarket, United Kingdom, **4** James Coburn AWCF Farriers Ltd, Newmarket, United Kingdom

☉ These authors contributed equally to this work.
‡ HC, DH, and LB also contributed equally to this work. KH and TP are senior authors on this work.
* khoran@rvc.ac.uk

**Data Availability Statement:** The data underlying the results presented in the study are submitted with this manuscript. Any questions pertaining to

## Abstract

Horseshoes influence how horses' hooves interact with different ground surfaces, during the impact, loading and push-off phases of a stride cycle. Consequently, they impact on the biomechanics of horses' proximal limb segments and upper body. By implication, different shoe and surface combinations could drive changes in the magnitude and stability of movement patterns in horse-jockey dyads. This study aimed to quantify centre of mass (COM) displacements in horse-jockey dyads galloping on turf and artificial tracks in four shoeing conditions: 1) aluminium; 2) barefoot; 3) GluShu; and 4) steel. Thirteen retired racehorses and two jockeys at the British Racing School were recruited for this intervention study. Tri-axial acceleration data were collected close to the COM for the horse (girth) and jockey (kidney-belt), using iPhones (Apple Inc.) equipped with an iOS app (SensorLog, sample rate = 50 Hz). Shoe-surface combinations were tested in a randomized order and horse-jockey pairings remained constant. Tri-axial acceleration data from gallop runs were filtered using bandpass Butterworth filters with cut-off frequencies of 15 Hz and 1 Hz, then integrated for displacement using Matlab. Peak displacement was assessed in both directions (positive 'maxima', negative 'minima') along the cranio-caudal (CC, positive = forwards), medio-lateral (ML, positive = right) and dorso-ventral (DV, positive = up) axes for all strides with frequency $\geq 2$ Hz (mean = 2.06 Hz). Linear mixed-models determined whether surfaces, shoes or shoe-surface interactions (fixed factors) significantly affected the displacement patterns observed, with day, run and horse-jockey pairs included as random factors; significance was set at $p < 0.05$. Data indicated that surface-type significantly affected peak COM displacements in all directions for the horse ($p < 0.0005$) and for all directions ($p \leq 0.008$) but forwards in the jockey. The largest differences were observed in the DV-axis, with an additional 5.7 mm and 2.5 mm of downwards displacement for the horse and jockey, respectively, on the artificial surface. Shoeing condition significantly affected all displacement parameters except ML-axis minima for the horse ($p \leq 0.007$), and all displacement

the data can be directed to the corresponding author (Kate Horan: khoran@rvc.ac.uk).

**Funding:** This study was achieved as a collaboration between the University of Edinburgh (KH MSc project) and a Royal Veterinary College project titled 'Shoe Assessment for Equine Racing' (KH postdoctoral research assistant), which is funded by the Horserace Betting Levy Board. The funder provided support in the form of salaries for authors [KH (full-time) and KK, JC, PD, HC, DH, LB and LH for data collection], but did not have any additional role in the study design, data collection and analysis, decision to publish, or preparation of the manuscript. The specific roles of these authors are articulated in the 'author contributions' section.

**Competing interests:** TP is the owner of EquiGait Ltd providing equine gait analysis products and services. JC is the owner of James Coburn AWCF Ltd, which employed JC, HC, DH and LB at the time of the study. JC, PD, HC and DH are currently registered farriers. This does not alter our adherence to PLOS ONE policies on sharing data and materials.

parameters for the jockey (p<0.0005). Absolute differences were again largest vertically, with notable similarities amongst displacements from barefoot and aluminium trials compared to GluShu and steel. Shoe-surface interactions affected all but CC-axis minima for the jockey (p≤0.002), but only the ML-axis minima and maxima and DV-axis maxima for the horse (p≤0.008). The results support the idea that hoof-surface interface interventions can significantly affect horse and jockey upper-body displacements. Greater sink of hooves on impact, combined with increased push-off during the propulsive phase, could explain the higher vertical displacements on the artificial track. Variations in distal limb mass associated with shoe-type may drive compensatory COM displacements to minimize the energetic cost of movement. The artificial surface and steel shoes provoked the least CC-axis movement of the jockey, so may promote greatest stability. However, differences between horse and jockey mean displacements indicated DV-axis and CC-axis offsets with compensatory increases and decreases, suggesting the dyad might operate within displacement limits to maintain stability. Further work is needed to relate COM displacements to hoof kinematics and to determine whether there is an optimum configuration of COM displacement to optimise performance and minimise injury.

## Introduction

Horseracing is a high-profile and high-risk sport, in which optimising the safety alongside performance of participating horses and jockeys is paramount. The risk of catastrophic musculoskeletal injuries in Thoroughbred flat races is 0.80 per 1000 race starts in the United Kingdom (UK), which compares to a global incidence of 1.17 per 1000 [1]. In addition, racehorse musculoskeletal injury accounts for up to 82% of lost training days lost in the UK [2], with 40% of two-year old horses in race training sustaining a musculoskeletal injury [3]. Extensive coverage of horseracing in the media means these injuries often draw the attention of the public globally, as well as the regulators; for example, the recent high injury rates at racecourses in California [4]. Failure to train and race due to injury also has a significant and detrimental economic impact. Consequently, there is increasing emphasis on proactive interventions to improve training and racing conditions, and prevent injury [5].

Maintaining a harmonious interaction between horse and jockey is one key aspect influencing safety, as biomechanical instabilities are the trigger behind most horse falls and jockey injuries [6]. In racing, a jockey positions themselves off the saddle in a two-point seat and their leg joints flex and extend in a rhythmical manner that aligns with the vertical oscillations of their horse's trunk [7]. Their body moves only a small amplitude with respect to a world inertial frame and is decoupled from the movements of the horse [8]. The horses' limbs act in sequence to redirect their centre of mass (COM) [9]. The first footfalls (hindlimbs) accelerate the horse, propelling the COM forwards, and the later ones (forelimbs) decelerate the horse and apply vertical impulse to the COM [10]. Energy is lost during the stance phase of the limbs and is a function of the change in the angle of the COM trajectory [9]. The leading forelimb is thought to be the most important for redirecting the COM, as a result of cranio-caudal (CC) deceleration, vertical acceleration and an increase in potential energy of the COM occurring during the stance phase of this limb [11]. On a stride per stride basis, jockey kinematics adjust to accommodate the changes in translational and rotational upper-body movements of the horse and thereby maintain stability [12]. For example, during stance of the leading hindlimb at gallop, the horse's trunk and jockey's pelvis both displace laterally and roll away from the side of this leg [12]. Force data from stirrups indicate that jockeys push away from the stirrup

on the non-lead side at this time to maintain the position of their COM close to the horse's midline and balance themselves [12]. A slight delay in the dorso-ventral (DV) displacement of a jockey's pelvis occurs relative to the horse, but medio-lateral (ML) movements are in phase [12]. Importantly, as the jockey's COM moves out of phase with the horse in the CC-axis [8, 12], this is when they are most unstable [12]. It is not known at which point in the stride cycle jockeys are most susceptible to injury, but it is plausible that it would be when displacements peak in the CC-axis.

Although the riding technique employed in racing is metabolically and mechanically costly for the jockey [13–15], increased inertia is detrimental to athletic performance [16] and this technique attenuates the acceleration and deceleration of the jockey in each stride cycle, with respect to a world reference frame. Consequently, it decreases energy expenditure for the horse and can optimize racing speed [8]. In trotting horses, it has been observed that a two-point seat also transfers the lowest and most constant load to a horse's back [17]. A potential drawback of the two-point seat position is the reduced points of horse-rider contact; a jockey only physically connects with their horse via the reins and their legs. This limits opportunity for tactile information exchange, which is known to play an important role in some equestrian disciplines [18, 19]. However, in racing, a jockey may signal desired alterations to their horse's biomechanical output by shifting their COM; "rider urging" has been linked to reduced stride length and increased stride frequency [20]. Adjustments to jockey position may additionally influence the patterns and symmetry of horses' movement, as has been documented in other disciplines [21–23].

To date, objective quantification of horse-rider interactions has tended to focus on equestrian disciplines in which the highest level of coupling between horse and rider movements is desired, for example in dressage [24–27]. Two studies have also explored horse-rider interactions during endurance riding [28, 29], and one study has investigated these in a racing context, using five horses [8]. Data indicate that both gait and ridden seating style are key influences on horse-rider movement patterns and their stability over time. However, there has been little research investigating how extrinsic factors, such as ground surface conditions or farriery interventions, might further influence these interactions, despite the potential implications for performance and injury risk. The interaction between hooves and the surfaces they are galloping over is at the heart of the risk of slippage, fractures and falls. If different properties of the hoof-surface interface alter traction on impact, limb loading rates and push-off forces, then the timing and magnitude—in 3D space—of horse and rider movements could change. Horse falls result from a temporary instability of the horse to support its COM or a failure of the musculoskeletal system due to injury. Jockey injury is, in most cases, due to horse falls [6].

Several epidemiological studies have identified ground surface type as a significant risk factor for injuries to racehorses [e.g. 30–33]. In the UK, most horse races are run on turf but training takes place on both turf and artificial surfaces. Ideally, on landing a surface will permit toe penetration to produce low resistance to shear forces, while also providing enough resistance to support the foot with some slip [34]. Some degree of hoof slip is advantageous for lowering the forces during deceleration [35, 36] and reducing bending moments on the cannon bone [37]. However, excessive hoof slide can predispose to injury, such as tears to the digital flexor muscles [38]. Depending on how hoof-ground interactions influence rider motion, it is possible that the rider may exacerbate the influence of a surface on the loading environment of the equine limb. Equine hoof kinematics have also been related to dynamic surface properties, including hardness, cushioning, responsiveness, grip and uniformity [39]. For example, hoof deceleration on impact in galloping horses shows an inverse relationship with track rebound rate; an effect that allows a smoother transition from stance to propulsion and increases stride

efficiency [40]. Lower hoof vibrations, accelerations and ground reaction forces have been found on synthetic surfaces compared to dirt surfaces [41, 42]. In addition, lameness and injury incidents have been linked to the magnitude of impact forces [43, 44], with surface implicated as a trigger factor for altering superficial digital flexor tendon loading and joint kinematics [45].

The influence of shoes on horse kinematics and injury mechanics has received less attention than surfaces. Within racing, the use of horseshoes is tightly regulated in most countries, including the UK. During flat racing, the British Horseracing Authority generally enforces that horses are shod with raceplates [46]: the majority have aluminium raceplates all-round and a smaller number have light steel or a mixture of aluminium (front) and steel (hind); this selection is intended to reduce added load and drag at gallop [47]. In contrast, in non-racing disciplines novel horseshoe materials, styles and innovative shoeing techniques are adopted more rapidly and are successfully used for many horses to influence biomechanical output. Perhaps in racing, horse and jockey safety needs to be considered more carefully before allowing new shoes due to the particularly high risk of injury [48], not only through falls but also through flying horseshoes at high speeds. Epidemiological data suggest certain shoe-types, such as those used in the US with high toe grabs, rims or pads, are associated with a higher risk of racehorse injury [49–52]. Hoof conformation also appears relevant, with long toes and low heels being linked to a greater risk of injury, particularly to the flexor tendons and suspensory ligament [52–54]. Nevertheless, a lack of study on novel versus existing shoe-types means jockeys, horse owners, farriers and veterinarians have been cautious to move away from traditional types, namely aluminium and light-steel. It appears short-sighted not to explore the potential benefits of new shoe and surface combinations.

Outside of racing, plastic shoes and pads made of synthetic rubber have been found to significantly reduce decelerative force and vibration frequency relative to steel shoes on concrete [55–57]. Steel shoes also impose higher maximal vertical forces compared to a barefoot condition [58]. Differences in vertical and breaking forces are likely to alter loading patterns, and thereby alter the duration over which push-up forces are transferred to a rider. Shoe shape is also relevant: for example, rolled toes might smooth breakover [59], eggbar shoes may shift the centre of foot pressure caudally [60] and solar protrusions, such as studs, may alter the balance between slip and grip [61]. Shoe mass is potentially important too, given the alterations to upper-body kinematics imposed on show-jumping horses with weighted boots [62]. Equine locomotor biomechanics, including foot-surface interaction are also influenced by conformation. This means data from horses used in other equestrian disciplines may not be directly transferable to the Thoroughbred racehorse, which is expected to perform on different surfaces at higher speeds and with a typically flat-foot, low heel conformation [63].

This study sought to assess how variations in hoof-ground interaction, imposed by different shoe and surface conditions, influence the COM displacements in galloping racehorse-jockey dyads. We hypothesised that horse movements in 3D space would differ between gallop trials in different shoe and surface conditions and that these differences would be mirrored in jockey movements. We expected movements in the DV and CC axes to show the greatest differences amongst shoe-surface conditions, owing to variability in hoof impact and push-off forces influencing the former and variations in slip duration influencing the latter.

## Materials & methods

### Ethics

Ethical approval for this intervention study was granted by The Royal (Dick) School of Veterinary Studies (R(D)SVS) Veterinary Ethical Review Committee (VERC, reference number

112:19) and the Human Ethical Review Committee (HERC, reference number 387–19) at the University of Edinburgh. The Royal Veterinary College (RVC) Clinical Research Ethical Review Board also approved the study (URN 2018 1841–2). Informed consent was given by the participating jockeys, farriers and horse owners. The individual in this manuscript has given written informed consent (as outlined in the PLOS consent form) to publish these case details.

## Data collection

**Horse-jockey dyads.**   Retired Thoroughbred ex-racehorses in regular work and utilised for jockey education at the British Racing School (BRS) in Newmarket, UK, provided a convenience sample of thirteen horses for this study. All horses were considered sound by the jockey, farrier and BRS management prior to data collection. They ranged in age from 6–20 years old and had body masses between 421 and 555 kg. Full details on body dimensions and hoof morphometrics are available in [64]. Two jockeys, both with >3 years of professional experience, were available for this study. All horses were ridden in a race exercise saddle. Jockey stirrup lengths varied between 47 and 50% of their leg lengths from the hip down.

**Farriery interventions.**   Each study horse had its hooves trimmed by a farrier according to a standardised trimming protocol to ensure consistent hoof geometry prior to data collection. This meant hoof geometry was always representative of the beginning of a trimming/shoeing cycle. The horses underwent data trials on artificial (Martin Collins Activ-Track) and turf surfaces in the following four shoeing conditions: 1) aluminium raceplates (Kerkhaert Aluminium Kings Super Sound horseshoes; 2) barefoot; 3) GluShus (aluminium-rubber composite horseshoes); and 4) steel shoes (Kerkhaert Steel Kings horseshoes) (S1 Table in S1 File). The horseshoe selection was decided upon after consulting farriers for their recommendations on existing and novel shoeing conditions to trial. The selection includes relevant and accessible options for racehorses in both training and racing contexts. The three shoe types were applied with five copper-coated mild steel nails. Shoe mass varied between 104–158 g (mean = 134±3g, mean ±2 s.e. unless otherwise stated) for the aluminium shoes (n = 63), 145–249 g (mean = 191 ±7 g) for the GluShus (n = 56), and 235–573 g (mean = 343±16 g) for the steel shoes (n = 65). The different shoe-surface combinations (Fig 1) were tested in a randomized order in case of carry-over effects between trials, for example due to tiredness of the horse or jockey.

**Measuring devices.**   The magnitude and variability of horse and jockey displacements were quantified non-invasively using two inertial measurement unit (IMU) devices. lMUs have been validated as a reliable and repeatable method for objectively quantifying equine locomotion [65–68] and human pelvic motion [69]. The IMU devices selected were iPhones equipped with the 'SensorLog' app (version 3.0, sample rate = 50 Hz). In alignment with previous studies on horse-rider interaction in dressage and endurance disciplines [25, 28, 29], we sought to fix these devices to the girth of the horse and the jockey's pelvis. These attachment locations permit approximate COM movement to be characterised [70, 71], without inhibiting the natural movement of the horse or jockey. For the horse, the iPhone was secured in a pouch on a stud girth (Fig 2a). The second iPhone for the jockey was placed in a neoprene kidney-belt (Fig 2b). The orientation of the iPhone on the girth and at the jockey's pelvis meant that at gallop the acceleration axes had the following orientations: in the CC direction positive was orientated forwards; in the ML direction positive was right; and in the DV direction positive was up (Fig 2c). For the CC-axis, 'forwards' versus 'backwards' movement was defined relative to the position that would be achieved at constant speed.

**Racing conditions.**   Following an initial ridden warm-up, each horse galloped on level (0–2% incline) artificial and turf tracks in each shoeing condition. The tracks curved slightly anti-clockwise (S1 Fig in S1 File). The jockey was asked to gallop their horse on both left and right

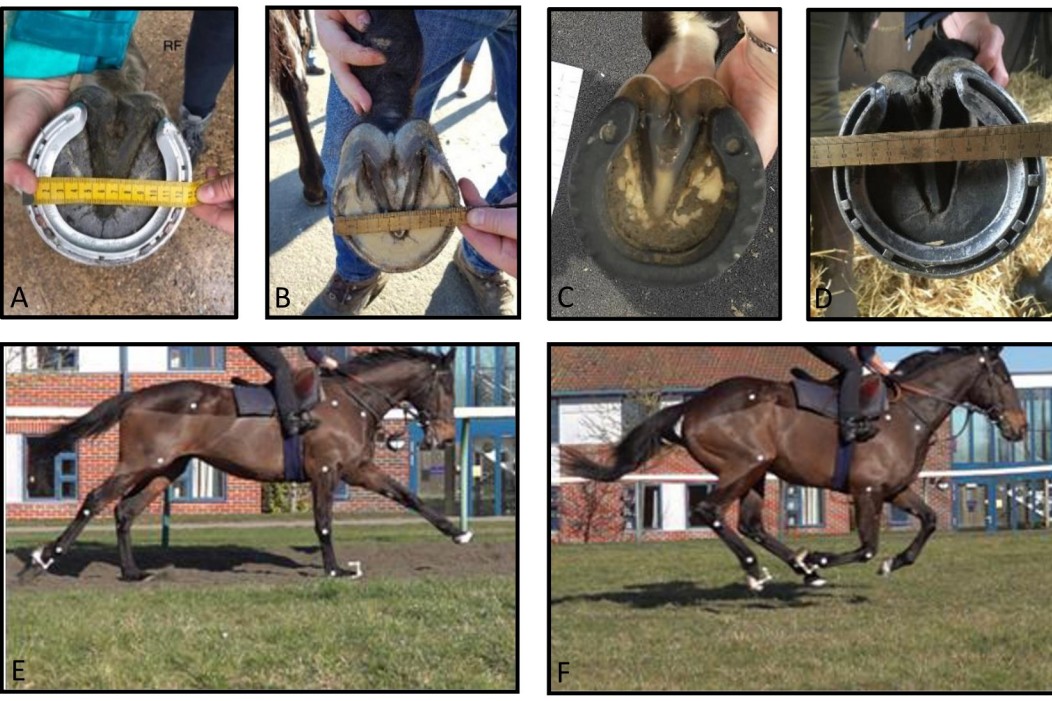

**Fig 1. Photographs of the four shoeing conditions and two surfaces used in this study. A)** Aluminium raceplate. **B)** Barefoot hoof. **C)** GluShu. **D)** Steel shoe. **E)** Horse galloping on Martin Collins Activ-Track at the British Racing School. **F)** Horse galloping on turf track at the British Racing School.

leads for each shoe-surface combination, in case of any laterality bias [72]. For some trials there were additional runs per condition, reflecting the fact that some additional equipment instrumented simultaneously required adjustments and repeat runs, as well as the need for multiple attempts for some horses to achieve the desired lead. Horses were not forced to exercise for a duration beyond what is typical of a short riding session (15–20 minutes), so trials were split across multiple days for each horse-jockey dyad. Data collection took place on the artificial track from summer 2019 through to early spring 2020. Data collection on the turf track was constrained to the mid-autumn of 2019 through to early spring of 2020, due to routine accessibility restrictions implemented by the BRS to avoid 'hard' going. Surface conditions for the turf track were documented by the jockeys, using terms used within the racing industry to describe "going" [73], and ranged from 'soft' to 'good-firm'. Further details on weather conditions on data collection days are available in [64].

## Data analysis

**Identifying gallop data, filtering and integration.** The approximate times of day for gallop episodes were noted during data collection and together with SensorLog speed data helped identify the relevant portions of accelerometry data. A custom-written Matlab script facilitated the extraction of 2–3 minute intervals of accelerometry data, each including a gallop episode, from the large original SensorLog 'csv' files. To reduce unwanted signal components and improve the overall precision of calculated parameters, data were filtered using a bandpass Butterworth filter with a passband of 1–15 Hz. Acceleration data were then integrated twice to quantify displacement following published methods [67, 68]. Through the highpass (bandpass) filter, the displacement had an average zero value.

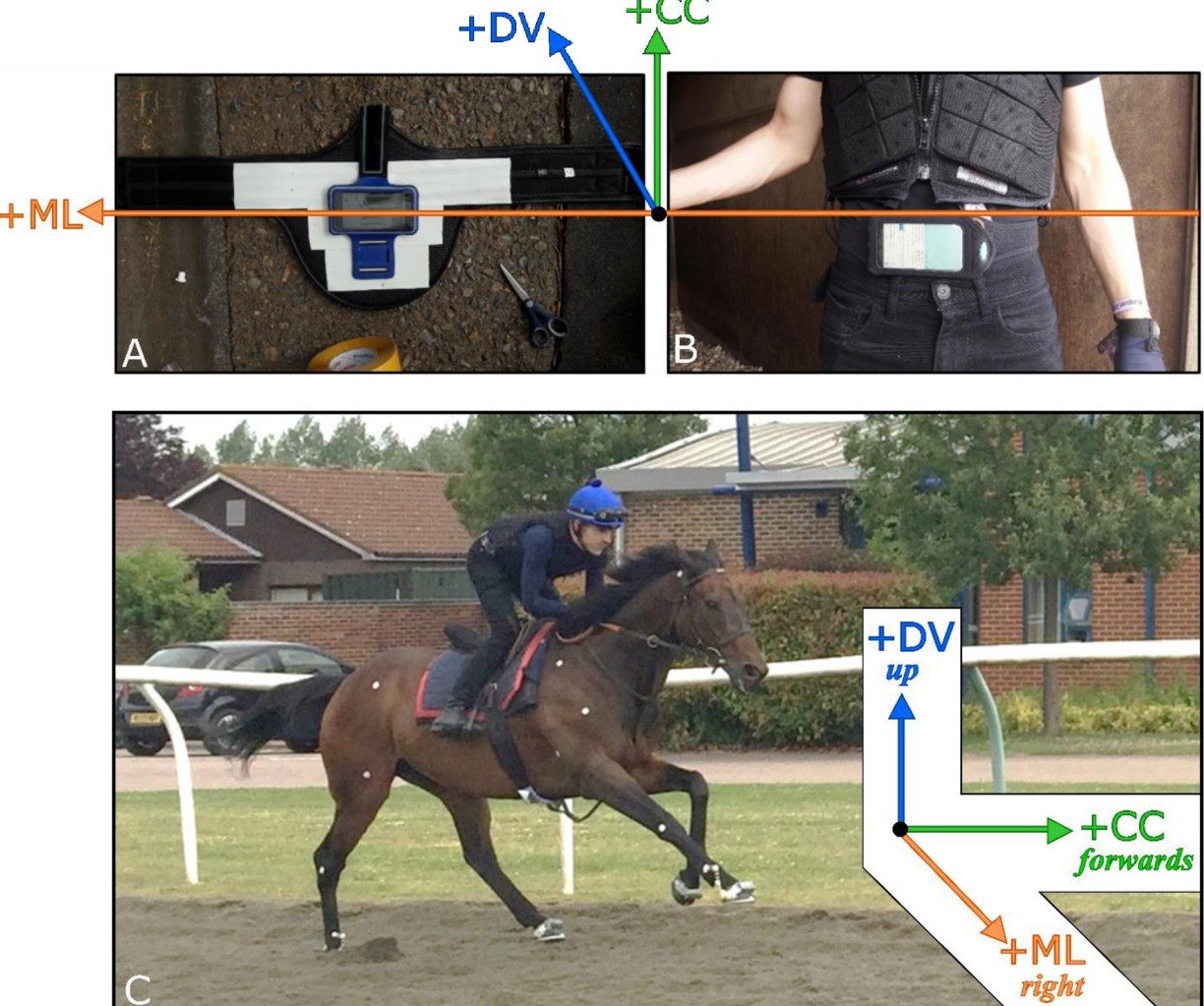

**Fig 2. Orientation of the cranio-caudal (CC), medio-lateral (ML) and dorso-ventral (DV) acceleration axes in the SensorLog app. A)** At the horse's girth. **B)** At the jockey's pelvis. **C)** At gallop, with the jockey in the two-point seat position.

**Quantifying horse-jockey interaction.**    Maximum displacements in positive ('maxima') and negative ('minima') directions were quantified along the CC, ML and DV axes and compared between horse and jockey in response to the different shoe and surface conditions. Due to restrictions of the SensorLog app and the phone operating system, horse and jockey sensor data could not be time-synchronized to the nearest millisecond. Hence, here data analysis was limited to movement amplitudes and the relative phase between horse and jockey displacement was not assessed. To account for the possible confounding effect of differences in speed on hoof kinematics [40], or limb coordination and inter-stride movement consistency [74], stride frequency was quantified during trials, as this provides an approximation of speed [75].

To quantify displacement minima and maxima, individual trial data were first segmented into strides in Microsoft Excel using a threshold based-method that identified corresponding points in the data sequences. A custom-written Matlab script was then used to further narrow down these data based on stride frequency. A frequency of $\geq 2$ Hz was used as the cut-off, which is approximately equivalent to 9 m s$^{-1}$ [75]. Data from the condensed files were plotted

so signal amplitude could be assessed visually to help identify the sequence of gallop strides. The magnitudes of displacement minima and maxima for gallop strides were quantified using Microsoft Excel. Wherever possible, consecutive strides were analysed but occasional skips of 2–3 strides reflected the horse galloping close to the 2 Hz cut-off frequency.

**Statistical testing.**   Statistical analyses were performed in SPSS. Twelve linear mixed models were used to test for significant differences in the magnitude of horse and jockey displacement minima and maxima in the CC, ML and DV axes, under the different shoe and surface conditions. Shoe, surface and 'shoe-surface interaction' were defined as fixed factors and horse-jockey pair, day and run numbers as random factors. Bonferroni post-hoc comparison tests were used to determine pairwise significance between shoes and shoe-surface combinations. The significance threshold in all statistical tests was set at $p < 0.05$.

The full data analysis procedure is summarized in Fig 3.

## Results

### Overview

From a total of 223 gallop runs, 185 horse and 187 jockey data files were viable for analysis. Runs were mostly discounted due to technical issues, but occasionally as a result of the horse not meeting trial criteria. Technical issues included the SensorLog app failing to record for a period of time, recording at a variable (and lower than expected) frequency or the phone unexpectedly losing power. If a horse bucked during a trial or ran with a stride frequency lower than the 2 Hz cut-off, then strides were not available for analysis. In addition, horse data from days 1 and 2 were discounted because the attachment of the phone to the girth was subsequently altered to improve security. No data from trials involving Horse-5 were viable. The total number of gallop runs analysed per shoe-surface condition is detailed in Table 1. Mean stride frequency across horse and jockey trials was 2.10±0.12 and 2.01±0.12 Hz (±1 s.d.), respectively, indicating stride frequency did not vary greatly beyond the 2 Hz threshold. Raw data are available in the S1 Data. Examples of acceleration data in time-series, integrated once to velocity and again to generate the required displacement data are available in the (S2, S3 Figs in S1 File) for both horse and jockey stride cycles.

### Displacement minima and maxima

Here, we first present the collective data of all shoe-surface conditions to assess the magnitude of the tri-axial displacements and look for general patterns in horse and jockey movements. We then explore the data in more detail to see whether shoe or surface had any additional impact on the results.

**General trends.**   Overall raw means for displacement along each axis from stride data collated across data trials are summarized in Table 2.

The position of the jockey relative to the horse throughout a gallop stride is illustrated in Fig 4, alongside an extract of displacement data for multiple strides. Jockey displacements were reduced relative to the horse in all directions. This difference was proportionally greatest in the CC-axis, particularly in the forwards direction (Table 2, Fig 4).

There was a relationship between the magnitude of displacement minima and maxima for both horse and jockey across individual trials (Fig 5). However, when displacement minima and maxima for the different displacement axes were compared, there appeared to be no clear relationship between them; Fig 6 illustrates a comparison between the DV and CC axes.

Normalizing CC-axis to DV-axis displacements for minima ('backwards/down') and maxima ('forwards/up') assisted in teasing apart these general patterns further (Fig 7). Although vertical displacements were almost always larger than CC displacements for both horse and

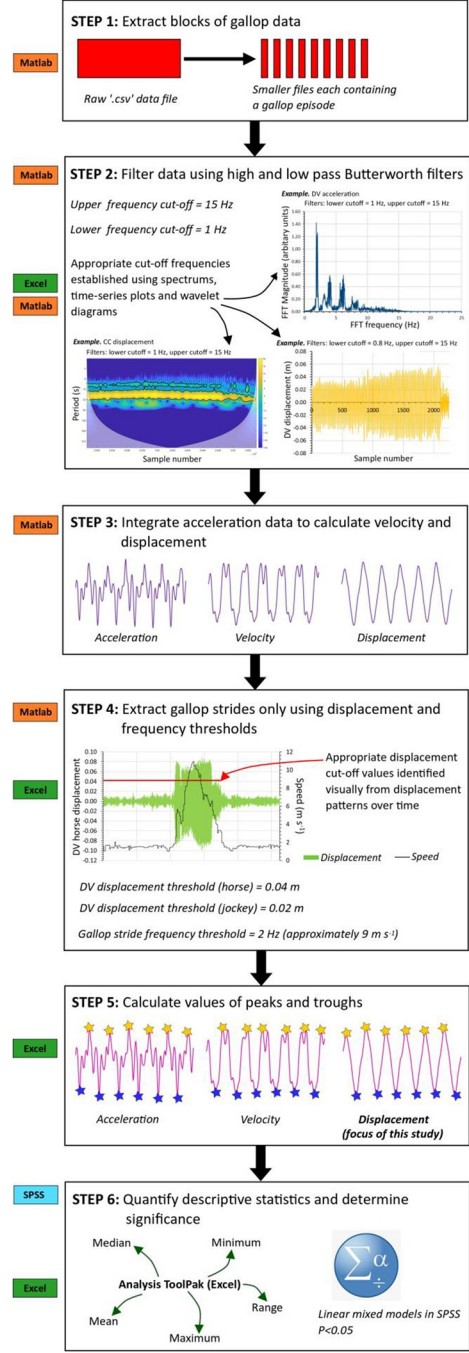

**Fig 3. Data analysis flowchart showing the step-by-step method used to quantify centre of mass displacement patterns for horse and jockey.** CC = cranio-caudal, DV = dorso-ventral.

**Table 1. Number of gallop runs analysed for each shoe-surface combination.**

| Shoe-surface combination | Number of trials | Number of viable horse data files | Number of viable jockey data files |
|---|---|---|---|
| Aluminium-Artificial | 32 | 19 | 19 |
| Aluminium-Turf | 21 | 19 | 19 |
| Barefoot-Artificial | 32 | 21 | 24 |
| Barefoot-Turf | 27 | 26 | 24 |
| GluShu-Artificial | 25 | 25 | 25 |
| GluShu-Turf | 28 | 25 | 25 |
| Steel-Artificial | 30 | 23 | 24 |
| Steel-Turf | 28 | 27 | 27 |

**Table 2. Mean displacement minima and maxima based on collated stride data.**

| Displacement parameter | Horse raw mean ±2 SD (mm) (n = 7525) | Jockey raw mean ±2 s.d. (mm) (n = 7695) |
|---|---|---|
| CC-axis minima (backwards) | 34 ± 16 | 16 ± 15 |
| ML-axis minima (left) | 13 ± 10 | 10 ± 11 |
| DV-axis minima (downwards) | 65 ± 26 | 46 ± 26 |
| CC-axis maxima (forwards) | 45 ± 21 | 17 ± 12 |
| ML-axis maxima (right) | 13 ± 11 | 11 ± 15 |
| DV-axis maxima (upwards) | 66 ± 16 | 39 ± 23 |

CC = cranio-caudal, ML = medio-lateral, DV = dorso-ventral.

jockey, there appeared to be a divergence in horse and jockey data as CC displacement magnitudes increased. In addition, jockey data were concentrated at lower CC/DV displacement ratios than the horse data, indicating that the jockeys experienced relatively more DV displacement.

**Influence of shoe and surface.** The distribution in raw mean displacement minima and maxima data for individual runs, sub-divided by shoe-surface combination, is presented in Fig 8.

Once individual stride data had been compiled across all trials for statistical testing, a total of 7525 and 7695 strides were available to assess horse and jockey displacements, respectively. Estimated marginal means (EMMs) computed by the models are provided in Tables 3–5 for surface, shoe and shoe-surface effects, respectively. Significance values for the main effects (surface, shoe) and the shoe-surface interaction are reported in Table 6. Significant differences amongst surfaces, shoes and surface-shoe combinations are indicated on Figs 9–11. A Bonferroni correction was applied to post-hoc pairwise comparisons of shoes and shoe-surface combinations (S2, S3 Tables in S1 File), and histograms confirmed normality of model residuals.

Surface significantly affected all displacement parameters for the horse ($p < 0.0005$) and all except the CC-axis maxima for the jockey ($p \leq 0.008$) (Fig 9). The largest differences were observed in the DV-axis; an additional 5.7 and 2.5 mm of downwards displacement on the artificial surface for the horse and jockey, respectively. Small increases in backwards displacement were also apparent for both horses (2.0 mm) and jockeys (0.6 mm) on turf. ML-axis differences were all $\leq 1.9$ mm, and the surface linked to the highest displacement was inconsistent between horses and jockeys.

Shoeing condition significantly affected all displacement parameters except the ML-axis minima for the horse ($p \leq 0.007$) and all displacement parameters for the jockey were

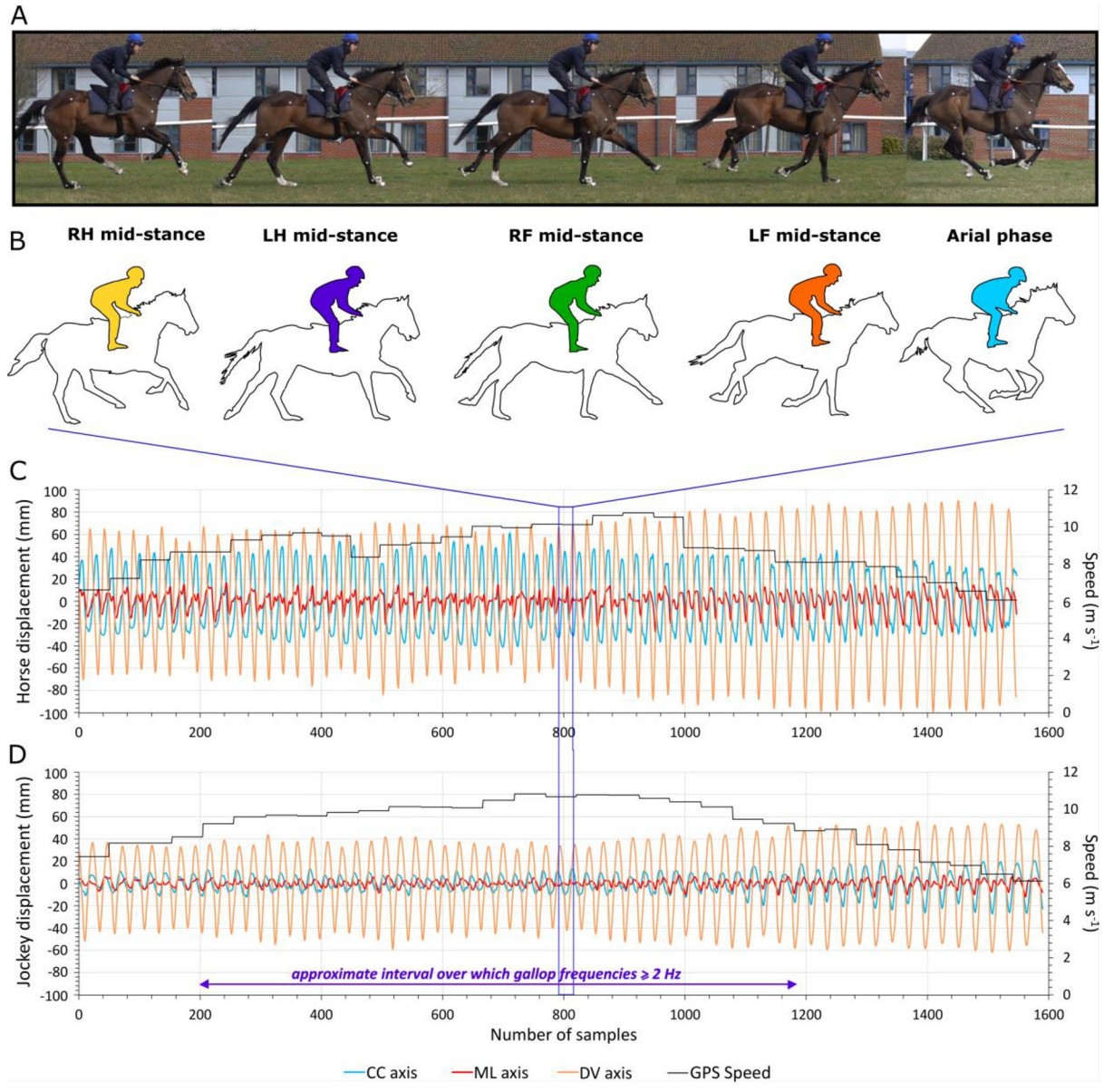

**Fig 4. Overview of horse and jockey displacements. A**. Photographs of horse and jockey over a gallop stride cycle illustrating changes in their body movements. **B**. Schematic diagrams highlighting the subtle variation in jockey position shown in A. **C**. Horse tri-axial displacements during a gallop interval. **D**. Jockey tri-axial displacements over the same gallop interval shown in C. Note that the alignment of speed data to acceleration data is only approximate.

significantly affected (p<0.0005) (Fig 10). Significance values for post-hoc pairwise comparisons between the four shoeing conditions are provided in the Supporting information for displacement parameters that were significantly affected (S2 Table in S1 File). The most notable difference for the horse occurred in DV-axis minima between the barefoot condition and steel shoes; there was an additional 2.9 mm of downwards displacement for barefoot. The ML-axis was least affected by shoeing condition. For the jockey, EMM displacement magnitude differences were largest for DV-axis minima (barefoot–steel = 3.1 mm) and maxima (aluminium–steel = 3.2 mm). For both horse and jockey, similarities in DV-axis data were apparent for barefoot and aluminium versus GluShu and steel.

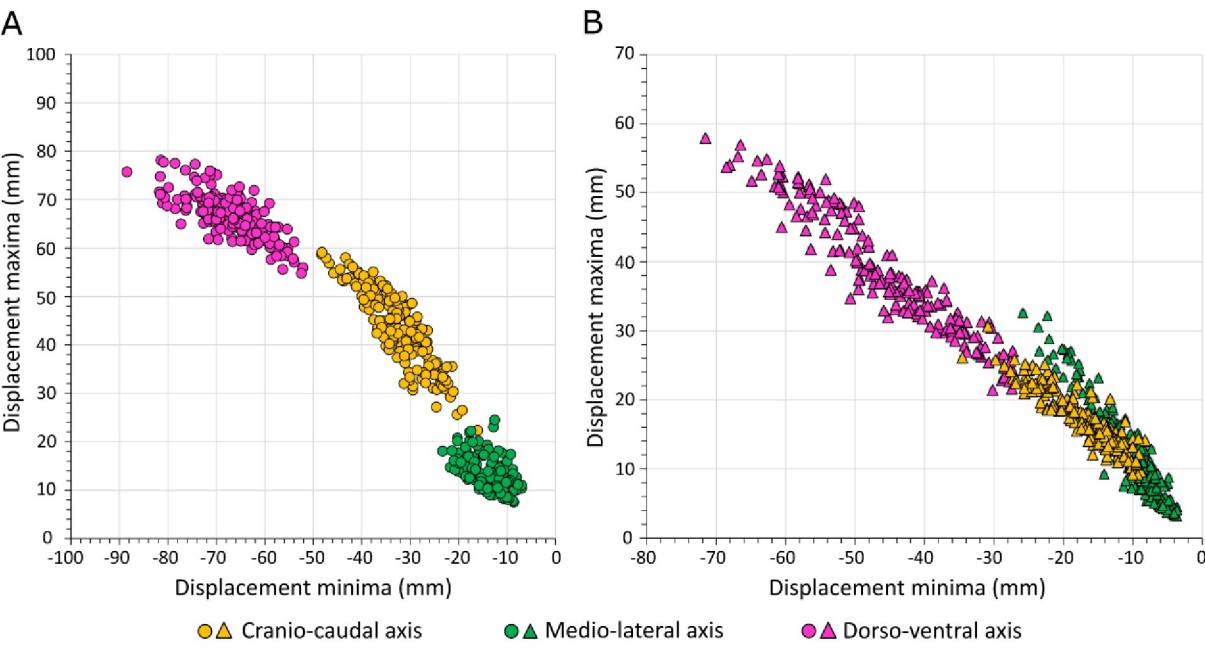

**Fig 5. Positive relationship between the magnitude of displacement minima and maxima for individual trials. A**. Horse data. **B**. Jockey data.

The interaction between shoeing condition and surface type significantly affected ML-axis minima, ML-axis maxima and DV-axis maxima data for the horse (p≤0.008), and all displacement parameters except CC-axis minima for the jockey (p≤0.002) (Fig 11). Post-hoc tests were performed when the interaction was significant. As there were eight possible shoe-surface combinations arising from two surfaces and four shoeing conditions, sequentially comparing these against one another gave rise to 28 comparisons per axis direction. With six possible directions reflecting movement in both positive and negative directions for CC, ML and DV displacement, there was a maximum of 168 comparisons possible. However, for the jockey, for whom shoe-surface interactions only significantly affected displacement in five of the six directions, 140 comparisons were made. Of these comparisons, the EMMs were significantly different in 83 cases. For the horse, 84 comparisons were made for the three directions that were significantly affected and 55 shoe-surface EMMs were found to be significantly different (S3 Table in S1 File). The largest differences in EMMs occurred in the vertical axis between 'barefoot-artificial' versus 'steel-turf': a combined minima and maxima average increase of 7.7 mm and 6.2 mm for 'barefoot-artificial' for the horse and jockey, respectively.

## Discussion

The aim of this study was to assess COM displacements in horse-jockey dyads galloping on turf and artificial surfaces in four shoeing conditions. Optimal stability in riding is often linked to synchronous horse-rider displacements [12], which may be best achieved when there is haptic communication between horse and rider via the saddle, such as in sitting trot and canter [25, 28, 29]. Therefore, the crouched position adopted by jockeys, with its isolated COM [8], is expected to be associated with reduced stability and an increased risk of falls [12]. It is important to establish how extrinsic factors modulate this risk. The findings of this study support the hypotheses that horseshoe and ground surface types can have significant effects on the upper body displacements of horses and their jockeys.

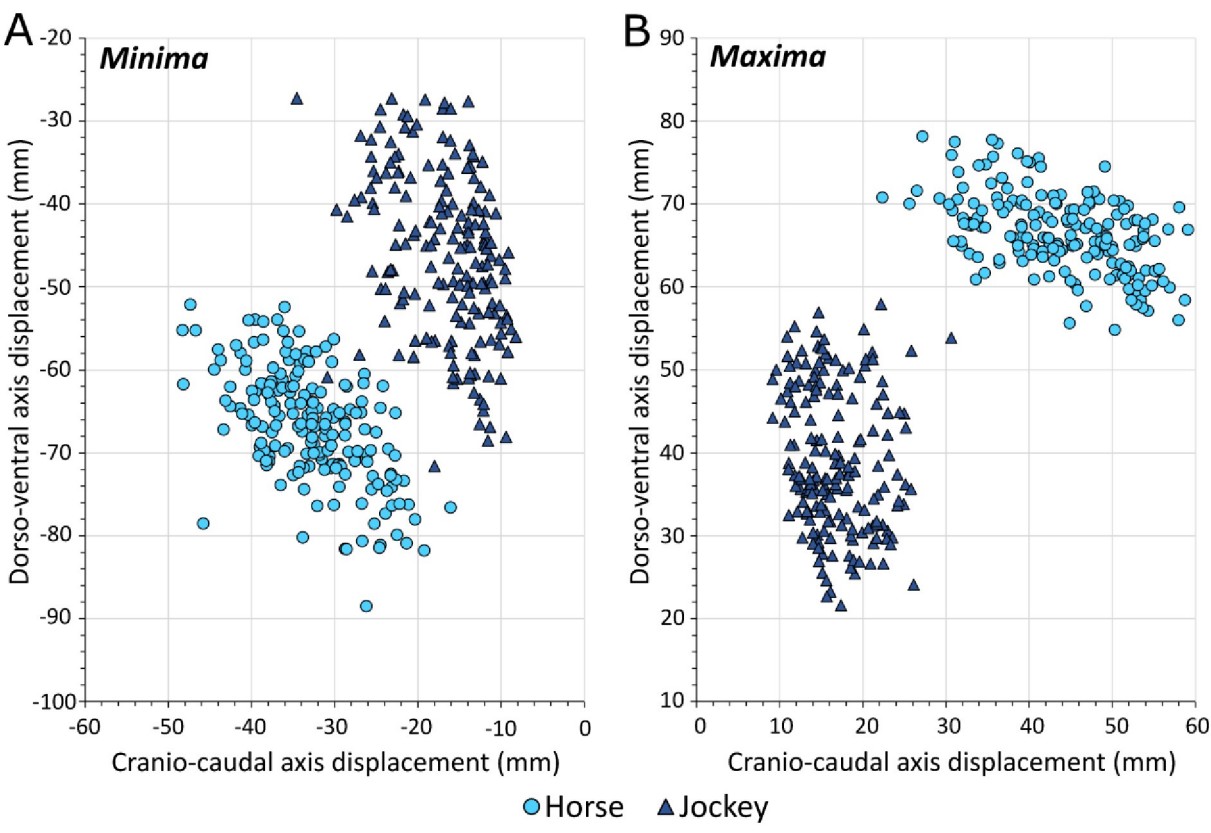

**Fig 6. Relationship between the magnitude of dorso-ventral (DV) axis and cranio-caudal (CC) axis displacements for horse and jockey individual trial data. A.** Minima. **B.** Maxima.

This study focused on quantifying minima and maxima of displacements over stride cycles, which is analogous to equine gait analysis systems that utilize vertical displacement minima and maxima to detect lameness [76]. However, shifts in the overall range of motion, rather than asymmetries, were of interest here. The stride frequency threshold, for data to be included in the analysis, is consistent with slow galloping speeds [77] but should exceed canter speeds of Thoroughbreds [78–80]. Broad-scale horse and jockey displacement magnitudes recorded are in alignment with previous work [8]. The general similarities in displacement minima and maxima magnitudes (Fig 5) reflect mostly symmetrical cyclical movements of the COM of each body. Greater consistency in this relationship for the jockey, may reflect fine motor control assisting balance; for example, jockeys may pull on the reins or lean on their horses' necks to improve CC balance, while lateral balance may be aided by sideways pushing of the legs, alterations to knee angles or changing the force distribution between stirrups [81]. This would lead to a 'smoother' more sinusoidal displacement curve with more similar values for minima and maxima. Interestingly, the horse CC displacement data showed higher maximum values for a given minimum value (Fig 5). These deviations from the general trend indicate that the CC displacement data do not perfectly follow a sine wave where the high-pass filtering would mean that the minima and maxima have very similar values. This may be linked to the fact that movement in the forwards direction was sharper, with high amplitude while movement in the backwards direction was smoother and prolonged, with a lower amplitude. In the ML-axis, it is possible that a 'curve effect' could explain why the jockey data showed ML maxima

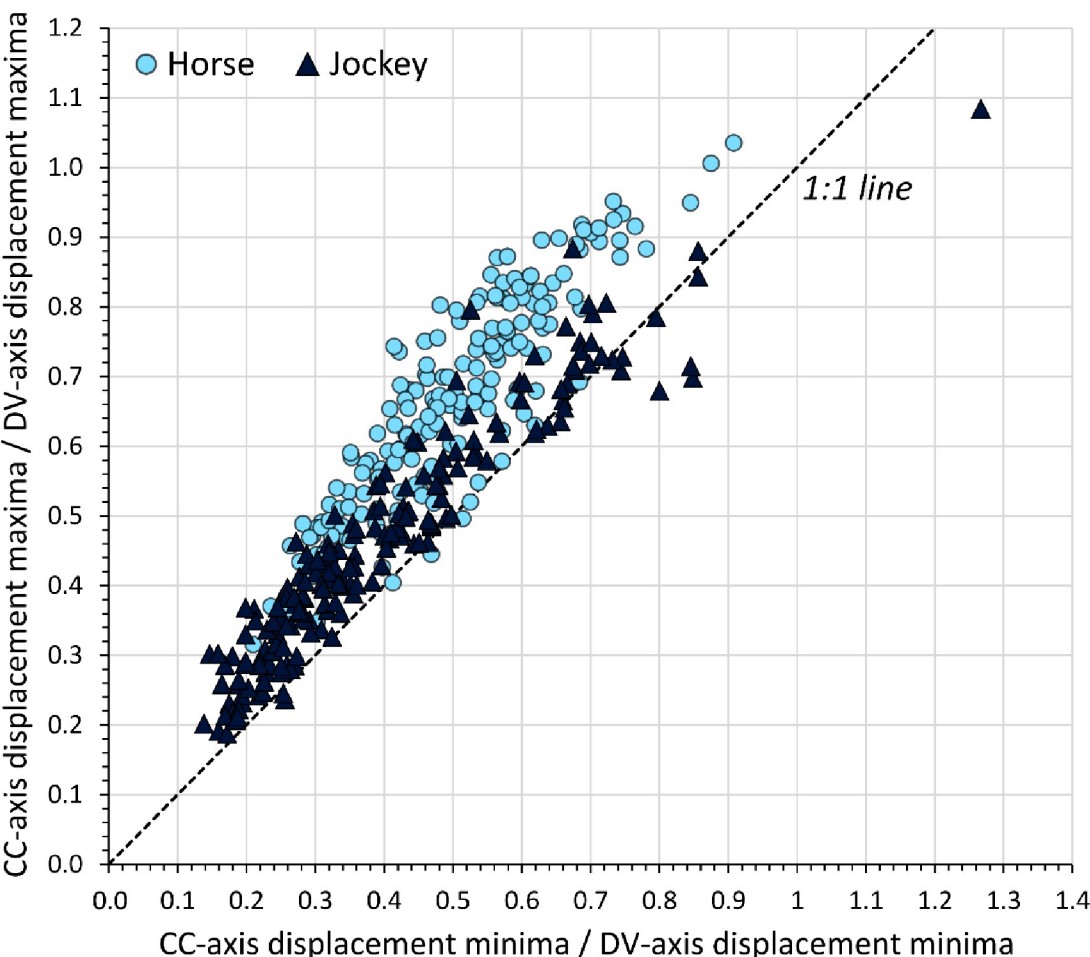

**Fig 7. Relationship between the ratio of cranio-caudal (CC) axis / dorso-ventral (DV) axis displacements for minima and maxima in horses and jockeys.** A 1:1 line is included for reference.

magnitudes that increased proportionally more with increasing values of ML minima, as the horses were galloping around a slight anticlockwise bend (S1 Fig in S1 File). Importantly, changes to the three-dimensional movement of the COM of the horse (and jockey) determines the distribution and magnitude of loads on limbs [82], and is therefore of relevance to injury.

The accuracy, precision and repeatability with current IMU sensor generations are in the order of 3–7 mm [83] and this is comparable between smartphones and specialist inertial sensing technology [67, 68, 84]. Here, many strides (7525 and 7695 for horse and jockey, respectively) from twelve horses were used to interpret the effect of different conditions. Although the differences between EMMs were small, the EMM values themselves exceeded sensor uncertainties. The smallest displacements occurred in the ML-axis, while larger displacements and displacement offsets occurred in the CC and DV axes; hence, the latter are more likely to be meaningful in a practical context. As the same horse-jockey combinations were assessed on the different surfaces and with different shoes we created a 'paired' comparison, which minimized the risk of individual horse (and jockey) related characteristics confounding results, such as skill or age.

Here, significantly greater vertical displacements were apparent for horses galloping on the artificial compared to the turf surface (Table 3, Fig 9). This may reflect the hooves sinking

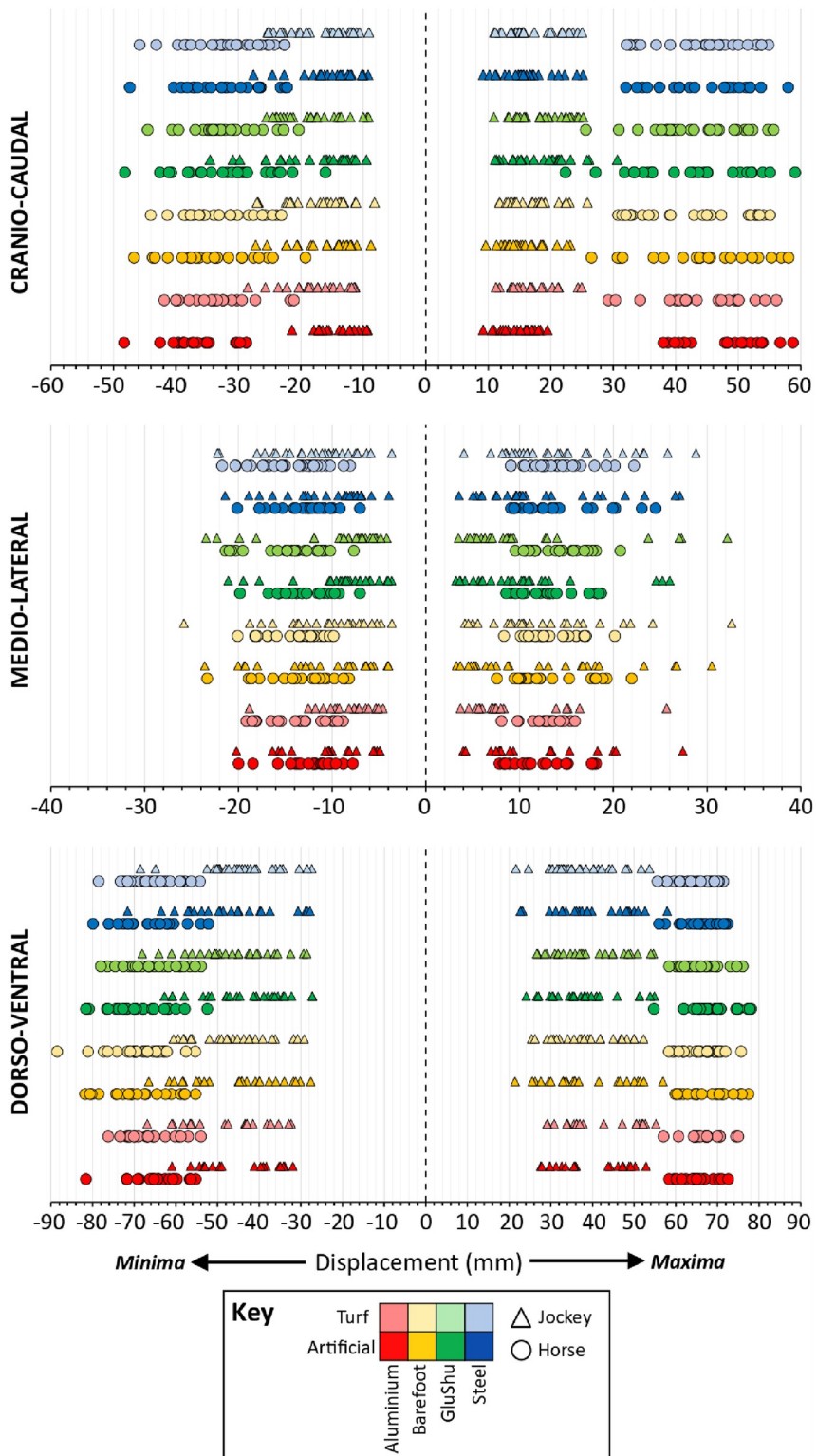

**Fig 8. Distribution of displacement minima and maxima values for horse and jockey, sub-divided by shoe-surface combination.** This is based on raw mean data of individual runs.

**Table 3. Estimated marginal means and confidence intervals for surface effects.**

| Displacement parameter | Surface | Horse Mean (mm) | Horse 95% Confidence Interval Lower Bound (mm) | Horse 95% Confidence Interval Upper Bound (mm) | Jockey Mean (mm) | Jockey 95% Confidence Interval Lower Bound (mm) | Jockey 95% Confidence Interval Upper Bound (mm) |
|---|---|---|---|---|---|---|---|
| CC-axis minima | Artificial | -32.39 | -35.84 | -28.95 | -16.94 | -19.90 | -13.98 |
| | Turf | -34.43 | -37.87 | -30.99 | -17.51 | -20.48 | -14.55 |
| ML-axis minima | Artificial | -12.97 | -14.18 | -11.76 | -11.17 | -15.84 | -6.51 |
| | Turf | -14.86 | -16.07 | -13.65 | -10.29 | -14.96 | -5.63 |
| DV-axis minima | Artificial | -69.07 | -73.07 | -65.08 | -44.18 | -49.93 | -38.43 |
| | Turf | -63.36 | -67.35 | -59.37 | -41.67 | -47.42 | -35.92 |
| CC-axis maxima | Artificial | 43.24 | 39.06 | 47.42 | 16.67 | 14.37 | 18.97 |
| | Turf | 45.35 | 41.17 | 49.53 | 17.00 | 14.70 | 19.30 |
| ML-axis maxima | Artificial | 12.65 | 10.75 | 14.56 | 13.61 | 7.82 | 19.41 |
| | Turf | 13.72 | 11.82 | 15.62 | 12.12 | 6.33 | 17.91 |
| DV-axis maxima | Artificial | 68.14 | 64.28 | 72.01 | 37.26 | 32.11 | 42.41 |
| | Turf | 64.62 | 60.75 | 68.48 | 35.86 | 30.71 | 41.01 |

CC = cranio-caudal, ML = medio-lateral, DV = dorso-ventral.

**Table 4. Estimated marginal means and confidence intervals for shoe effects.**

| Displacement parameter | Shoe | Horse Mean (mm) | Horse 95% Confidence Interval Lower Bound (mm) | Horse 95% Confidence Interval Upper Bound (mm) | Jockey Mean (mm) | Jockey 95% Confidence Interval Lower Bound (mm) | Jockey 95% Confidence Interval Upper Bound (mm) |
|---|---|---|---|---|---|---|---|
| CC-axis minima | Aluminium | -34.54 | -37.99 | -31.09 | -17.38 | -20.35 | -14.41 |
| | Barefoot | -33.45 | -36.90 | -30.01 | -17.51 | -20.48 | -14.55 |
| | GluShu | -32.85 | -36.30 | -29.39 | -17.61 | -20.59 | -14.64 |
| | Steel | -32.81 | -36.26 | -29.36 | -16.40 | -19.37 | -13.43 |
| ML-axis minima | Aluminium | -13.78 | -14.99 | -12.57 | -10.71 | -15.38 | -6.05 |
| | Barefoot | -14.00 | -15.21 | -12.79 | -10.63 | -15.30 | -5.97 |
| | GluShu | -13.75 | -14.97 | -12.53 | -10.37 | -15.04 | -5.70 |
| | Steel | -14.14 | -15.35 | -12.92 | -11.21 | -15.88 | -6.55 |
| DV-axis minima | Aluminium | -66.77 | -70.77 | -62.76 | -44.26 | -50.02 | -38.51 |
| | Barefoot | -67.73 | -71.73 | -63.73 | -44.51 | -50.26 | -38.76 |
| | GluShu | -65.56 | -69.58 | -61.54 | -41.47 | -47.24 | -35.70 |
| | Steel | -64.81 | -68.81 | -60.81 | -41.46 | -47.22 | -35.70 |
| CC-axis maxima | Aluminium | 45.55 | 41.36 | 49.73 | 16.80 | 14.49 | 19.10 |
| | Barefoot | 43.46 | 39.28 | 47.64 | 16.86 | 14.56 | 19.17 |
| | GluShu | 43.66 | 39.47 | 47.85 | 17.32 | 15.01 | 19.63 |
| | Steel | 44.52 | 40.34 | 48.71 | 16.36 | 14.06 | 18.67 |
| ML-axis maxima | Aluminium | 12.96 | 11.05 | 14.86 | 12.53 | 6.73 | 18.32 |
| | Barefoot | 13.14 | 11.24 | 15.04 | 12.25 | 6.45 | 18.04 |
| | GluShu | 13.10 | 11.19 | 15.01 | 13.09 | 7.29 | 18.89 |
| | Steel | 13.55 | 11.64 | 15.46 | 13.61 | 7.81 | 19.40 |
| DV-axis maxima | Aluminium | 67.25 | 63.38 | 71.12 | 38.29 | 33.13 | 43.44 |
| | Barefoot | 67.17 | 63.30 | 71.04 | 37.52 | 32.37 | 42.68 |
| | GluShu | 65.57 | 61.69 | 69.45 | 35.37 | 30.21 | 40.53 |
| | Steel | 65.54 | 61.67 | 69.41 | 35.07 | 29.91 | 40.22 |

CC = cranio-caudal, ML = medio-lateral, DV = dorso-ventral.

**Table 5. Estimated marginal means and confidence intervals for combined shoe-surface effects.**

| Displacement parameter | Surface | Shoe | Horse Mean (mm) | Horse 95% Confidence Interval Lower Bound (mm) | Horse 95% Confidence Interval Upper Bound (mm) | Jockey Mean (mm) | Jockey 95% Confidence Interval Lower Bound (mm) | Jockey 95% Confidence Interval Upper Bound (mm) |
|---|---|---|---|---|---|---|---|---|
| CC-axis minima | Artificial | Aluminium | -33.58 | -37.04 | -30.11 | -17.13 | -20.11 | -14.15 |
| | Artificial | Barefoot | -32.48 | -35.94 | -29.02 | -17.16 | -20.14 | -14.18 |
| | Artificial | GluShu | -31.71 | -35.17 | -28.24 | -17.42 | -20.41 | -14.43 |
| | Artificial | Steel | -31.81 | -35.28 | -28.35 | -16.05 | -19.04 | -13.07 |
| | Turf | Aluminium | -35.50 | -38.97 | -32.04 | -17.63 | -20.62 | -14.64 |
| | Turf | Barefoot | -34.43 | -37.89 | -30.97 | -17.87 | -20.85 | -14.88 |
| | Turf | GluShu | -33.99 | -37.45 | -30.52 | -17.81 | -20.79 | -14.82 |
| | Turf | Steel | -33.81 | -37.27 | -30.34 | -16.75 | -19.73 | -13.76 |
| ML-axis minima | Artificial | Aluminium | -12.73 | -13.96 | -11.50 | -11.79 | -16.46 | -7.12 |
| | Artificial | Barefoot | -13.38 | -14.61 | -12.15 | -10.73 | -15.40 | -6.06 |
| | Artificial | GluShu | -12.90 | -14.14 | -11.66 | -10.39 | -15.06 | -5.71 |
| | Artificial | Steel | -12.87 | -14.11 | -11.63 | -11.78 | -16.45 | -7.11 |
| | Turf | Aluminium | -14.83 | -16.07 | -13.58 | -9.64 | -14.31 | -4.97 |
| | Turf | Barefoot | -14.62 | -15.86 | -13.39 | -10.53 | -15.21 | -5.86 |
| | Turf | GluShu | -14.59 | -15.83 | -13.36 | -10.36 | -15.03 | -5.68 |
| | Turf | Steel | -15.40 | -16.63 | -14.17 | -10.65 | -15.32 | -5.98 |
| DV-axis minima | Artificial | Aluminium | -69.27 | -73.32 | -65.23 | -44.71 | -50.49 | -38.93 |
| | Artificial | Barefoot | -71.17 | -75.22 | -67.13 | -46.72 | -52.49 | -40.95 |
| | Artificial | GluShu | -68.19 | -72.25 | -64.13 | -41.69 | -47.48 | -35.91 |
| | Artificial | Steel | -67.66 | -71.72 | -63.60 | -43.61 | -49.39 | -37.83 |
| | Turf | Aluminium | -64.26 | -68.32 | -60.20 | -43.82 | -49.60 | -38.03 |
| | Turf | Barefoot | -64.28 | -68.33 | -60.24 | -42.29 | -48.07 | -36.52 |
| | Turf | GluShu | -62.93 | -66.98 | -58.88 | -41.25 | -47.03 | -35.46 |
| | Turf | Steel | -61.96 | -66.01 | -57.92 | -39.31 | -45.09 | -33.54 |
| CC-axis maxima | Artificial | Aluminium | 44.85 | 40.65 | 49.06 | 17.02 | 14.70 | 19.34 |
| | Artificial | Barefoot | 42.46 | 38.26 | 46.67 | 16.53 | 14.21 | 18.84 |
| | Artificial | GluShu | 42.20 | 37.99 | 46.41 | 17.09 | 14.77 | 19.42 |
| | Artificial | Steel | 43.44 | 39.23 | 47.65 | 16.05 | 13.73 | 18.37 |
| | Turf | Aluminium | 46.24 | 42.03 | 50.45 | 16.58 | 14.25 | 18.90 |
| | Turf | Barefoot | 44.46 | 40.25 | 48.66 | 17.20 | 14.88 | 19.52 |
| | Turf | GluShu | 45.12 | 40.91 | 49.33 | 17.54 | 15.22 | 19.86 |
| | Turf | Steel | 45.60 | 41.39 | 49.80 | 16.67 | 14.36 | 18.99 |
| ML-axis maxima | Artificial | Aluminium | 12.20 | 10.28 | 14.12 | 13.98 | 8.18 | 19.78 |
| | Artificial | Barefoot | 12.72 | 10.80 | 14.64 | 12.77 | 6.97 | 18.57 |
| | Artificial | GluShu | 12.41 | 10.48 | 14.33 | 13.36 | 7.56 | 19.17 |
| | Artificial | Steel | 13.29 | 11.37 | 15.21 | 14.34 | 8.54 | 20.14 |
| | Turf | Aluminium | 13.71 | 11.79 | 15.64 | 11.08 | 5.28 | 16.88 |
| | Turf | Barefoot | 13.56 | 11.64 | 15.48 | 11.72 | 5.92 | 17.52 |
| | Turf | GluShu | 13.79 | 11.87 | 15.71 | 12.81 | 7.01 | 18.61 |
| | Turf | Steel | 13.81 | 11.89 | 15.73 | 12.87 | 7.07 | 18.67 |

*(Continued)*

**Table 5.** (Continued)

| Displacement parameter | Surface | Shoe | Horse Mean (mm) | Horse 95% Confidence Interval Lower Bound (mm) | Horse 95% Confidence Interval Upper Bound (mm) | Jockey Mean (mm) | Jockey 95% Confidence Interval Lower Bound (mm) | Jockey 95% Confidence Interval Upper Bound (mm) |
|---|---|---|---|---|---|---|---|---|
| DV-axis maxima | Artificial | Aluminium | 68.41 | 64.52 | 72.30 | 38.16 | 32.99 | 43.32 |
| | Artificial | Barefoot | 69.70 | 65.81 | 73.59 | 38.73 | 33.56 | 43.89 |
| | Artificial | GluShu | 66.90 | 63.01 | 70.79 | 35.75 | 30.58 | 40.93 |
| | Artificial | Steel | 67.56 | 63.67 | 71.46 | 36.42 | 31.25 | 41.59 |
| | Turf | Aluminium | 66.09 | 62.19 | 69.98 | 38.42 | 33.25 | 43.59 |
| | Turf | Barefoot | 64.63 | 60.74 | 68.52 | 36.32 | 31.15 | 41.49 |
| | Turf | GluShu | 64.24 | 60.35 | 68.13 | 34.98 | 29.81 | 40.15 |
| | Turf | Steel | 63.51 | 59.62 | 67.40 | 33.71 | 28.55 | 38.88 |

CC = cranio-caudal, ML = medio-lateral, DV = dorso-ventral.

more at impact on the artificial surface, combined with greater propulsive forces being generated during the latter part of stance. Artificial surfaces may undergo a high degree of elastic deformation [85] and therefore return more energy to the hoof. A more rapid hoof breakover on the artificial surface [86] would be consistent with the notion that push-off forces, which would impose vertical COM movement, were stronger on this track, relative to turf. However, significantly higher CC horse displacement occurred on turf (Table 3, Fig 9). This could be linked to increased grip at the hoof-surface interface, as this would increase CC forces [56], and may suggest that surface traction had the dominant control on CC movement rather than any horizontal propulsive forces generated during breakover. Nevertheless, further work is needed to establish which surface conferred the greatest grip, as reduced breakover times on

**Table 6. Summary of output from linear mixed models: F and significance (p) values.**

| Displacement parameter | Source | F value (Horse) | Significance (Horse) | F value (Jockey) | Significance (Jockey) |
|---|---|---|---|---|---|
| CC-axis minima | Surface | 78.007 | <0.0005 | 7.111 | 0.008 |
| | Shoe | 21.849 | <0.0005 | 12.672 | <0.0005 |
| | Surface * Shoe | 0.266 | 0.850 | 0.278 | 0.842 |
| ML-axis minima | Surface | 125.766 | <0.0005 | 42.774 | <0.0005 |
| | Shoe | 2.054 | 0.104 | 12.159 | <0.0005 |
| | Surface * Shoe | 6.071 | <0.0005 | 28.665 | <0.0005 |
| DV-axis minima | Surface | 167.959 | <0.0005 | 53.192 | <0.0005 |
| | Shoe | 16.321 | <0.0005 | 35.530 | <0.0005 |
| | Surface * Shoe | 2.097 | 0.098 | 19.712 | <0.0005 |
| CC-axis maxima | Surface | 51.581 | <0.0005 | 3.623 | 0.057 |
| | Shoe | 22.820 | <0.0005 | 8.599 | <0.0005 |
| | Surface * Shoe | 2.524 | 0.056 | 4.801 | 0.002 |
| ML-axis maxima | Surface | 38.549 | <0.0005 | 74.127 | <0.0005 |
| | Shoe | 4.072 | 0.007 | 21.577 | <0.0005 |
| | Surface * Shoe | 3.989 | 0.008 | 17.793 | <0.0005 |
| DV-axis maxima | Surface | 179.093 | <0.0005 | 25.833 | <0.0005 |
| | Shoe | 21.925 | <0.0005 | 48.447 | <0.0005 |
| | Surface * Shoe | 13.428 | <0.0005 | 12.959 | <0.0005 |

Significant effects are highlighted in bold. CC = cranio-caudal, ML = medio-lateral, DV = dorso-ventral.

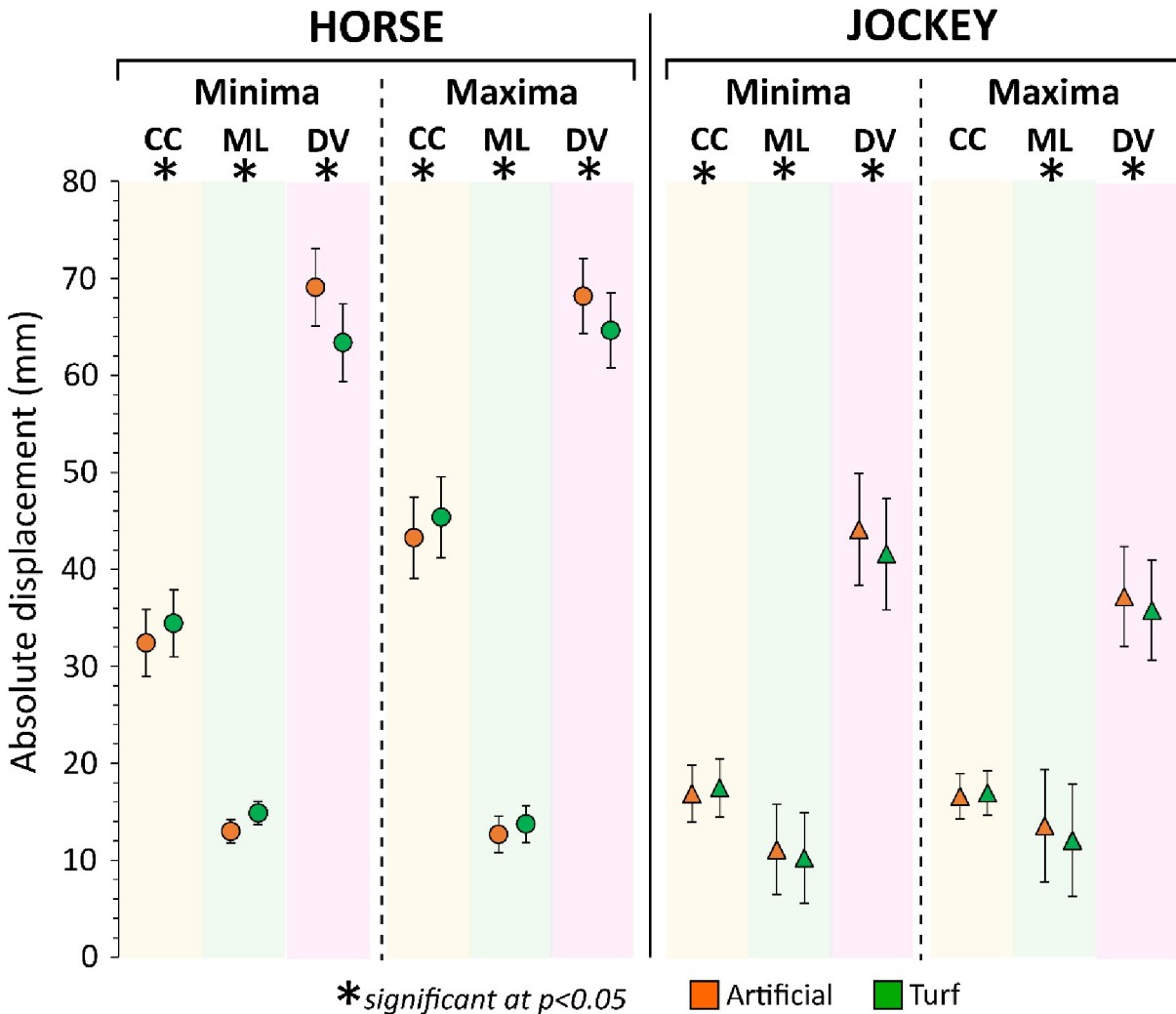

**Fig 9. Output of linear mixed models for surface effects.** Estimated marginal means with 95% confidence interval are presented per displacement parameter. CC = cranio-caudal, ML = medio-lateral, DV = dorso-ventral.

the artificial track could suggest the hooves had greater purchase on this surface [86]. It is also plausible that the softer nature of the artificial track caused the toe to rotate into this surface more easily than on turf [86]; this effect may help explain both why hoof breakover took longer on the turf track and why the associated upper body movements were more pronounced in the CC direction on turf. The patterns in horse displacements were mirrored in jockey displacements. It is important to consider that greater CC movement of the jockey could compromise stability. This is because their COM moves out of phase with the horse in this axis, in contrast to the DV and ML axes where horse and jockey movements are almost in phase [8, 12]. As such, jockey stability could be considered to be greater on the artificial surface, where CC movements were lower compared to turf, under the ground conditions studied. Stability is likely to be associated with safety, and jockey perception of safety and grip are strongly correlated [64]. It is therefore interesting to note that jockeys perceived grip to be enhanced on the artificial surface [64], which may suggest that they attribute the experience of lowered CC movement to increased grip at the hoof-surface interface.

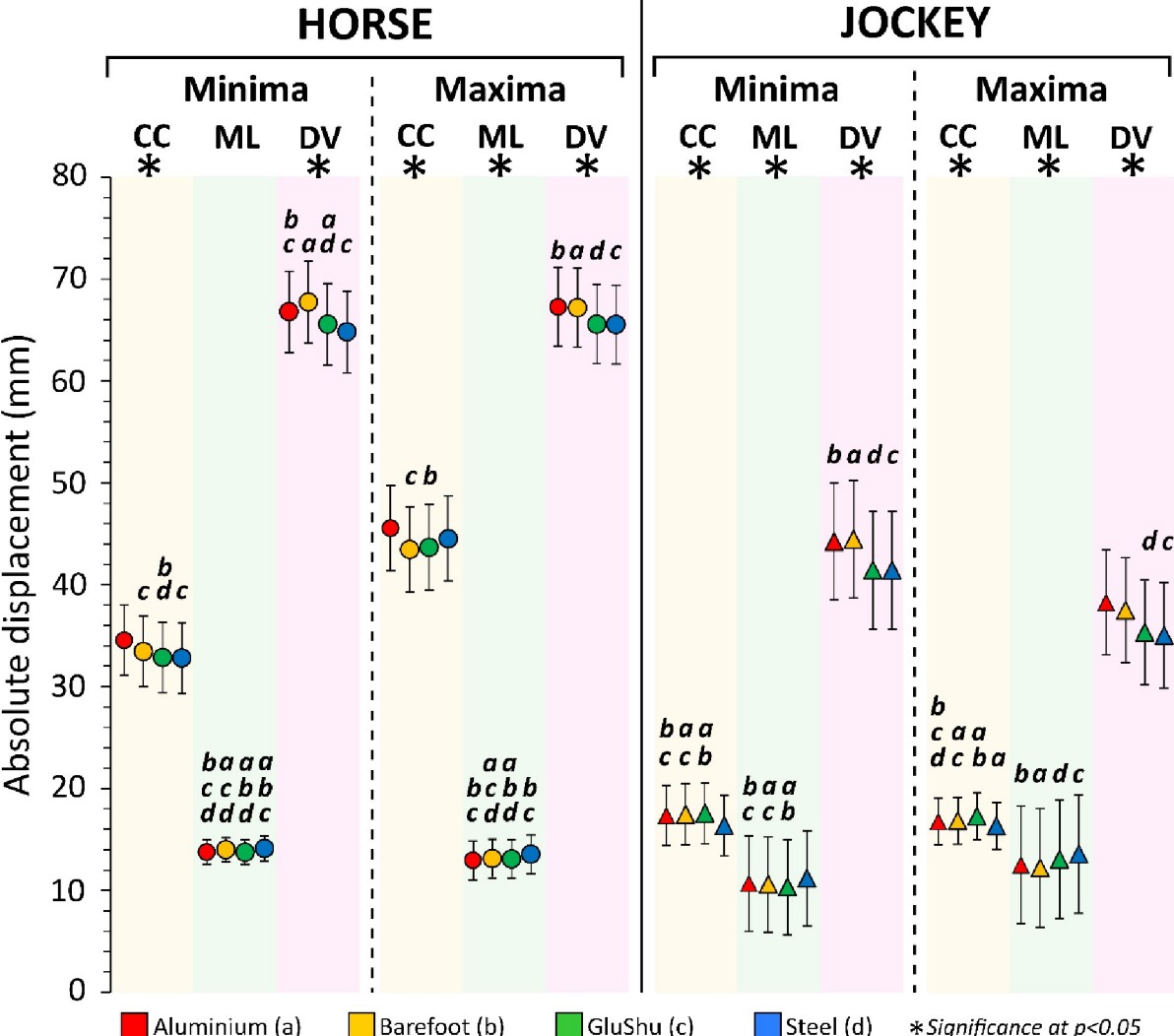

For each displacement parameter, shoe-types that do not share a subscript have significantly different estimated marginal means.

**Fig 10. Output of linear mixed models for shoe effects.** Estimated marginal means with 95% confidence interval are presented per displacement parameter. Significant differences between shoeing conditions detected in post-hoc tests (with Bonferroni correction) are indicated. CC = cranio-caudal, ML = medio-lateral, DV = dorso-ventral.

Amongst the shoe effects (Fig 10), vertical displacements were notably similar for barefoot and aluminium compared to GluShu and steel conditions for horse and jockey. Perhaps the additional distal mass, associated with steel and GluShus, increased the energetic cost of loco-motion for the horse [87] and led to an associated reduction in the vertical range of motion at the girth. Jockey displacements were most reduced in the CC-axis (0.4–1.2 mm) and increased in the ML-axis (0.5–1.4 mm) when riding horses in steel shoes, compared to all other shoe-types (Table 4). This may reflect a further adaptation of the jockey to counter energetic costs, since reduced CC displacement reflects reduced acceleration and deceleration of jockey COM per stride cycle [8]. This reduction in CC displacement may also indicate that jockey stability was improved under the steel shoeing condition. In general, the jockeys showed proportionally more vertical than horizontal displacement, but as overall horse-jockey CC displacements

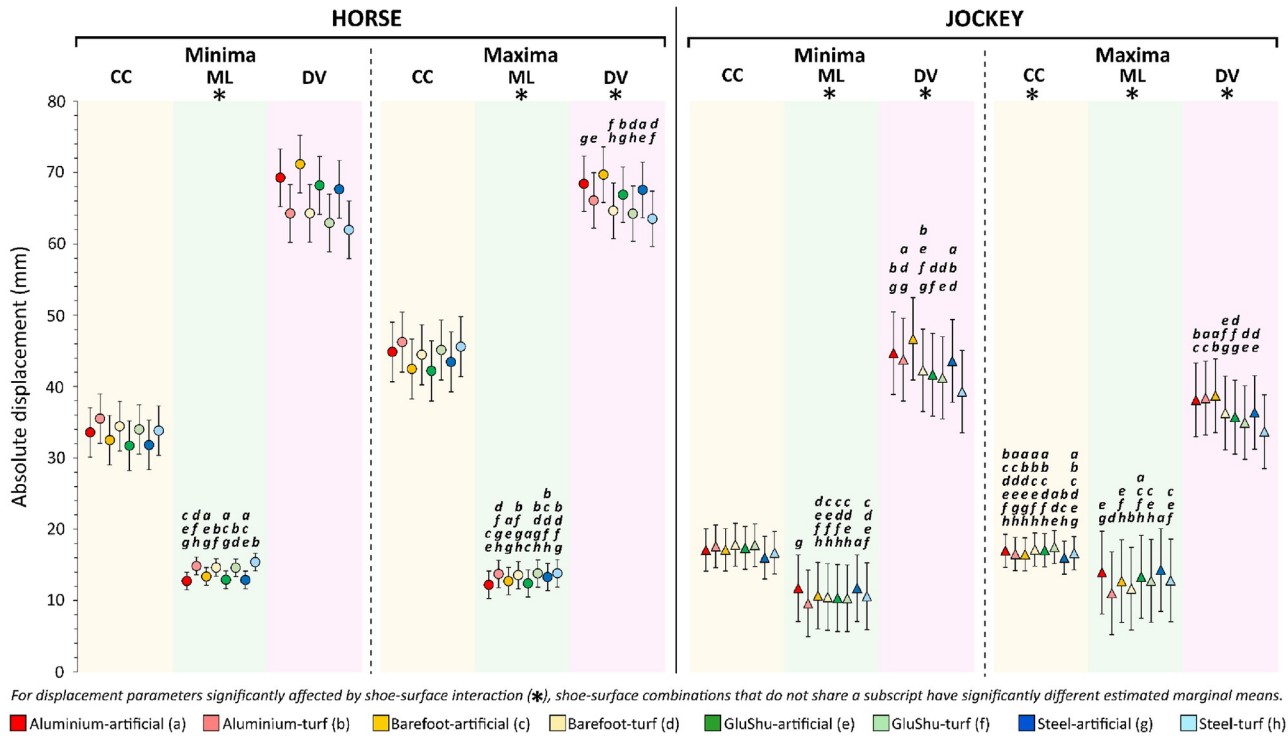

**Fig 11. Output of linear mixed models for surface-shoe combinations.** EMMs with 95% confidence intervals are presented per shoe-surface combination for each displacement parameter. Significant differences detected in post-hoc tests (with Bonferroni correction) are indicated. Note that pairwise comparisons between shoe-surface combinations in a particular axis direction were only made when there was found to be a significant interaction between shoeing condition and surface. CC = cranio-caudal, ML = medio-lateral, DV = dorso-ventral.

increased they appeared to move increasingly more backwards and/or upwards relative to the horses (Fig 7).

The largest absolute increase in EMMs amongst the shoe-surface conditions was observed in DV-axis minima for 'barefoot-artificial' compared to 'steel-turf', with 9.2 and 7.4 mm of additional upwards movement for horse and jockey, respectively. This may reflect the most extreme differences in distal-limb mass and material properties between shoes and surfaces. Specifically, the rigidity and relative hardness of steel [88], combined with potentially less responsive turf, may have driven rapid energy loss through limb vibrations and inelastic surface compaction; thus energy available for return to the horse would be reduced [39], when compared to 'barefoot-artificial'. Interestingly, horse-jockey displacements under the 'barefoot-turf' condition were not the most extreme. This might suggest that the 'barefoot-turf' condition did not present any greater movement challenge to the dyad from the perspective of relative COM displacements. However, current racing guidelines advocate to avoid this condition [46] and further work is needed to relate the COM data to injury risk. For example, it is possible that barefoot horses could be more susceptible to excessive slip and injury on heavy ground, which was not studied. It is a limitation of this study that ground conditions were not well-characterised on data collection days, as surface properties can be readily altered by water content and temperature [89–93].

Some inconsistencies in the significantly affected displacement parameters for horse and jockey could reflect non-harmonious interactions between the two bodies, or controlled interventions of the jockey aimed at altering gait or improving their own stability. Horses may also alter their gait in response to changes in grip characteristics of shoes to maintain a constant

slip time and distance [56]. Adjustments to joint angles, joint angular velocities and foot velocity at impact have been reported in response to altered shoeing conditions [94]. Riding experience and skill may influence rider ability to follow such subtleties in the movement patterns of their horse [95, 96], or to affect changes in gait by controlling their mass distribution, as has been reported in lameness examinations [97]. Jockeys with more experience are reported to have a lower risk of falling, and their horses have a lower risk of fatal limb fractures compared to jockeys with less experience [98–100]. This may be because advanced riders anticipate horse movement at a neuromuscular level with more defined and relaxed contralateral muscular activation patterns than novice riders [101, 102]. The jockeys in this study only had a limited number of professional years of experience, so their movement patterns may not be directly transferable to those of a more experienced jockey. Pain or stiffness can also limit rider ability to follow horse movements [18, 103] and the natural biomechanical asymmetries of the jockeys involved here was not assessed. However, since the horse-jockey pairs remained fixed it is not expected that these factors will have skewed our interpretation of horse and jockey response to the different shoe-surface combinations, although it is possible that the nuances of the response may differ amongst jockeys. It was impossible to blind the jockeys to the specific shoe-surface conditions being trialed; hence, they may also have introduced, consciously or unconsciously, alterations to their ridden style to accommodate anticipated differences. It is possible that directed jockey interventions explain why shoe-surface interactions more commonly affected jockey displacements (S3 Table in S1 File).

Thoroughbred racehorses show a variety of head, withers and pelvic movement asymmetries that vary in magnitude and direction [104, 105]. Some of these may result in changes to hoof landing angle, centre of pressure, and the distribution and timings of associated forces [106], with implication for the forces transferred to the jockey. Individual horse training experience may also bias horse-jockey interactions and horse adaptability to ridden exercise. For example, increases in stride frequency and protraction time over a six month training interval have been documented in racing Thoroughbreds [107]. Horse joint kinematics also vary with age: for example, maturity and increasing stiffness of the suspensory apparatus tissues in older Thoroughbreds lessens dorsi-flexion of forelimb fetlock joints [108, 109]. Previous research thus alludes to the importance of race and training interventions, workload and musculoskeletal health for altering horse-jockey movement dynamics.

The difference between horse and jockey EMMs for each shoe-surface combination for displacement minima and maxima were correlated (Fig 12), which reflects a link between the magnitudes of their cyclical movements. Overall differences in the CC-axis and DV-axis were 21.8 ±3.0 mm and 26.6±1.9 mm (mean ±2 s.e), respectively, and these have more tangible relevance than those in the ML-axis, which were considerably smaller (1.8±1.1 mm). Interestingly, however, the most extreme EMM difference in the DV-axis corresponded to the lowest EMM difference in the CC-axis (GluShu-artificial) and vice versa (aluminium-turf). There was also a general separation of artificial and turf data, with the former associated with relatively larger CC-axis and smaller DV-axis displacement offsets. Potentially, these trends signify a feedback loop was at play, whereby if a shoe-surface combination triggered horse-jockey displacement differences beyond a certain threshold in the CC-axis, then compensatory changes were required in the DV-axis for the dyad to maintain stability over a stride cycle. The correlation between horse-jockey EMM difference in the DV-axis versus the CC-axis was significant for minima displacements (Fig 13), suggesting it may have been easier to make adjustments in the backwards and downwards directions. Therefore, although there were no clear relationships between CC-axis and DV-axis data for the individual horse and jockey bodies (Fig 6), there may be value to interpreting their displacements collectively. Unfortunately, it was not possible to identify whether the horses or jockeys made the proposed adaptations instantaneously or over time.

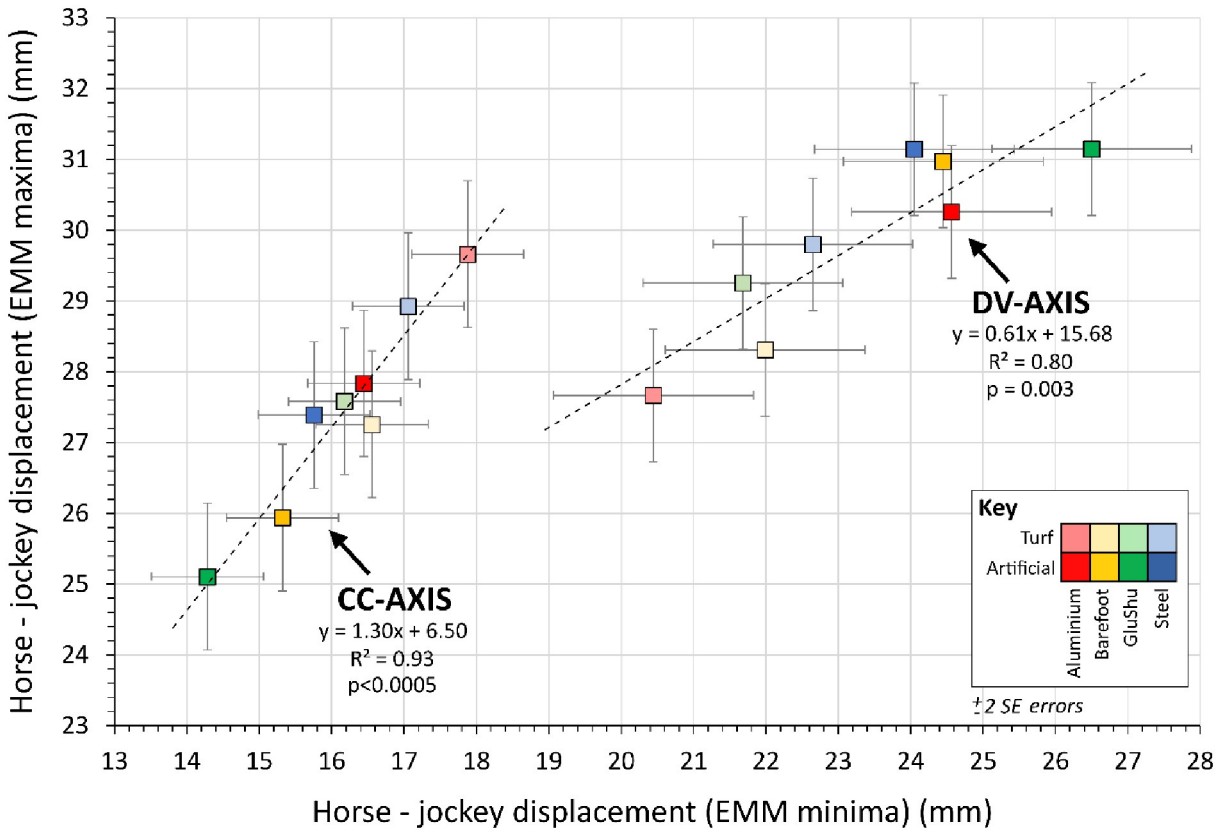

**Fig 12. Relationship between minima and maxima of horse-jockey estimated marginal mean (EMM) differences per shoe-surface combination, sub-divided by displacement axis.** Properties of the linear regression lines are indicated. Errors are ±2 SE. Note: horse-jockey EMM differences in the medio-lateral axis were considerably smaller (Table 5), so are not shown.

In this study, IMUs were constrained to two anatomical locations: horse girth and jockey pelvis. Although girth attachments may be well-suited to picking up broad-scale equine gait differences [25, 28, 29], it is possible that subtle changes imposed by shoes and surfaces may be masked at this location, or subject to inconsistencies if the jockey tightens the girth asymmetrically. In future, a multi-sensor system may enable displacement variability to be identified at certain anatomical locations that is not present elsewhere [110–112], and offer finer insights into horse-rider movements. It may also help pinpoint which locations provide the most useful information. Certainly at lower speeds, more comprehensive full-body IMU systems may be safe and achievable, and offer more detailed insights into horse-rider movement dynamics [111]. Determining the optimum balance of horse and jockey displacement parameters is challenging. However, it may be aided by linking the objectively measurable characteristics of COM displacements recorded here to: 1) measures of hoof kinematics, such as slip, impact, breakover velocity and push-off forces; and 2) the subjective assessment of jockeys, regarding how safe they felt and any ridden strategies they may have tried to implement under different shoe-surface conditions. Equipped with this knowledge, race-training programmes may be able to offer jockeys feedback on how they should adjust their body movements to optimise potential trade-offs between locomotion efficiency and safety. In addition, policy decisions regarding permitted shoe-surface combinations may be objectively supported or refuted. In future, it will also be helpful to consider displacement patterns throughout the entire stride cycle to assess in detail how COM displacements relate to each hoof-surface contact. This is

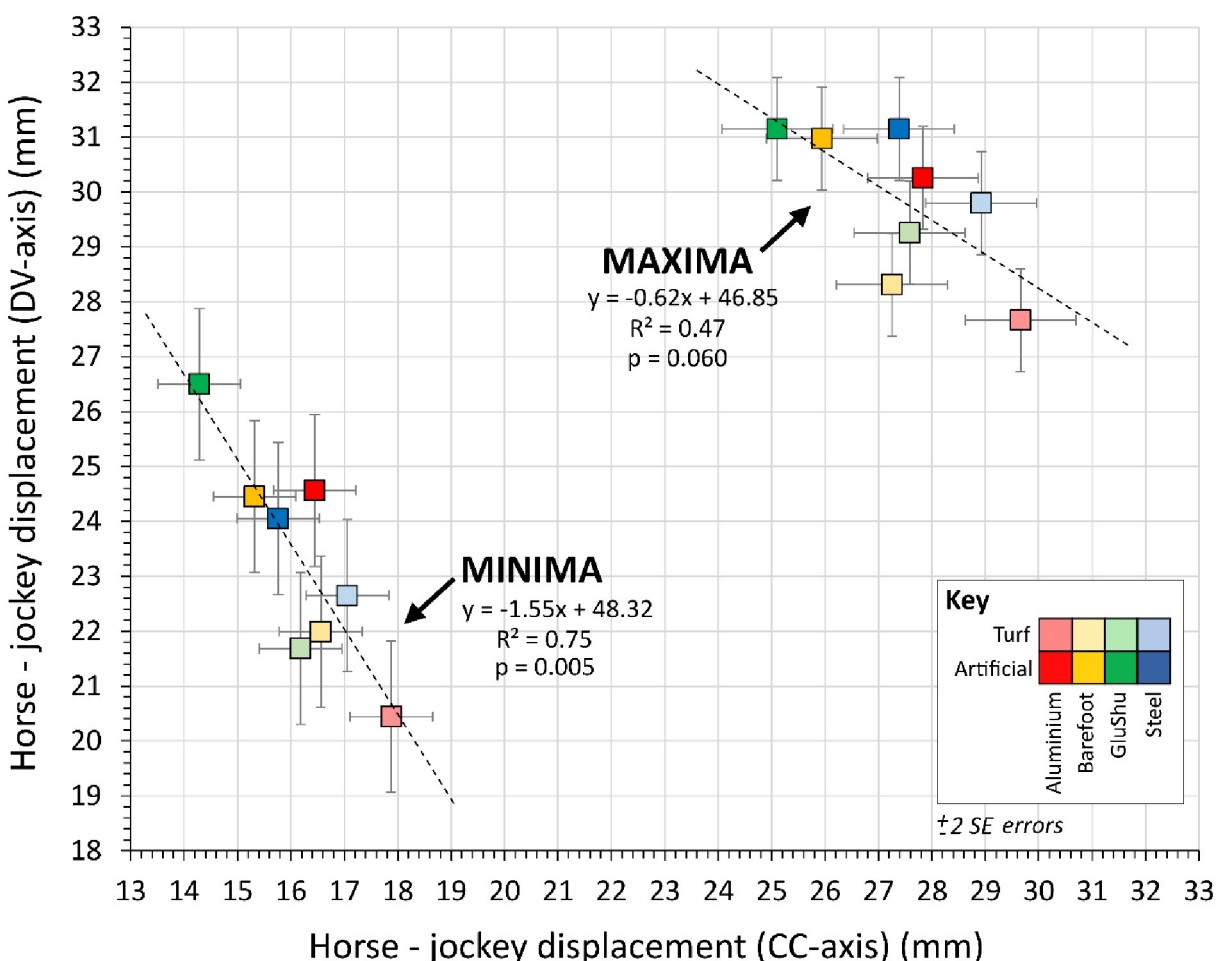

**Fig 13. Relationship between cranio-caudal (CC) axis and dorso-ventral (DV) axis data for horse-jockey estimated marginal mean (EMM) differences per shoe-surface combination, sub-divided into minima and maxima.** Properties of the linear regression lines are indicated. Errors are ±2 SE.

likely to be particularly insightful if time-synchronised horse-jockey displacement data become available, enabling horse-jockey relative phase offsets to be quantified also.

Nevertheless, it will be important to take a holistic approach to optimizing safety, performance and well-being, combining knowledge of intrinsic and extrinsic factors. Collection of physiological data from horse-jockey dyads, such as heart rate, lactate concentrations and cortisol concentrations, could be incorporated in future studies of this nature, to compare physiological responses to biomechanical output. In addition, understanding how different shoe-surface combinations damp impact vibrations is relevant for musculoskeletal health [91, 113–115]. It would also be interesting to explore horse-jockey responses in uphill or downhill runs when stride timing variables and peak vertical forces differ [116], as well as on turns or jumps. The current project could also be adapted to horses and riders in other equestrian disciplines, such as dressage, with horses working in different shoes or over different surfaces.

## Conclusion

This study demonstrated that surfaces and horseshoes can have significant effects on COM displacements of horses and jockeys at gallop. These effects were detected using IMU devices

fitted to the girth of horses and pelvis of jockeys, which recorded tri-axial acceleration. Displacement was calculated by integrating acceleration data. An increase in DV displacements on the artificial surface relative to turf may reflect greater hoof sink on impact followed by increased push-off. Higher cranio-caudal movements on turf could indicate that this surface afforded more grip under the ground conditions studied. More similar changes in DV displacement under aluminum and barefoot shoeing conditions when compared to GluShu and steel may reflect differences in shoe mass. The magnitude of horse minus jockey displacement in the CC and DV axes appeared to show compensatory increases and decreases, which may signify collective biomechanical adaptations of the dyad to maintain stability. Future work seeks to use multi-sensor IMU systems to determine which anatomical locations are most sensitive to variability at the hoof-surface interface and to relate COM movement dynamics to hoof kinematic variables, such as impact accelerations and slip distance. Ultimately, gaining a better understanding of the effect of novel and existing horseshoe-surface combinations on horse-jockey interactions at gallop is relevant for optimising performance, welfare and safety during both training and racing. It may offer opportunities to become prophylactic with regards to reducing the risk of falls, improving horse comfort and preventing catastrophic injuries in equine athletes and their jockeys.

## Supporting information

**S1 File. Supplementary methods and supplementary results.**
(DOCX)

**S1 Data. Raw data.**
(XLSX)

## Acknowledgments

We would like to thank the British Racing School for facilitating access to horses, riders and facilities. Jessica Josephson, Alice Morrell and Morgan Ruble from the Royal Veterinary College are thanked for assisting with data collection. Charlotte Woolley from the University of Edinburgh is also thanked for her support.

## Author Contributions

**Conceptualization:** Kate Horan, Thilo Pfau.

**Formal analysis:** Kate Horan.

**Funding acquisition:** Thilo Pfau.

**Investigation:** Kate Horan, Kieran Kourdache, James Coburn, Peter Day, Henry Carnall, Dan Harborne, Liam Brinkley, Lucy Hammond, Sean Millard.

**Methodology:** Kate Horan, Thilo Pfau.

**Project administration:** Kate Horan.

**Resources:** Kate Horan, Thilo Pfau.

**Supervision:** Bryony Lancaster, Thilo Pfau.

**Validation:** Kate Horan.

**Visualization:** Kate Horan.

**Writing – original draft:** Kate Horan.

**Writing – review & editing:** Kate Horan, Thilo Pfau.

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
