## [Decision Letter · Decision Letter 0]

11 Jun 2021

PONE-D-21-05125

The effect of horseshoes and surfaces on horse and rider centre of mass displacements at gallop

PLOS ONE

Dear Dr. Horan,

Thank you for submitting your manuscript to PLOS ONE. After careful consideration, we feel that it has merit but does not fully meet PLOS ONE’s publication criteria as it currently stands. Therefore, we invite you to submit a revised version of the manuscript that addresses the points raised during the review process.

We look forward to receiving your revised manuscript.

Kind regards,

Chris Rogers

Academic Editor

PLOS ONE

Journal Requirements:

'I have read the journal's policy and the authors of this manuscript have the following competing interests: Thilo Pfau is owner of EquiGait Ltd providing equine gait analysis products and services.'

We note that one or more of the authors are employed by a commercial company: EquiGait Ltd and James Coburn AWCF Ltd.

3. We note that Figure 2 includes an image of a [patient / participant / in the study]. 

Additional Editor Comments (if provided):

Thank you for your submission. Both reviewers have identified a number of minor editorial revision are required before acceptance.

Reviewers' comments:

Reviewer's Responses to Questions

**Comments to the Author**

1. Is the manuscript technically sound, and do the data support the conclusions?

Reviewer #1: Yes

Reviewer #2: Yes

2. Has the statistical analysis been performed appropriately and rigorously? 

Reviewer #1: Yes

Reviewer #2: Yes

3. Have the authors made all data underlying the findings in their manuscript fully available?

Reviewer #1: Yes

Reviewer #2: Yes

4. Is the manuscript presented in an intelligible fashion and written in standard English?

Reviewer #1: Yes

Reviewer #2: Yes

5. Review Comments to the Author

Reviewer #1: A well-presented and really interesting applied study that provides some preliminary insight into the impact of surfaces and shoe types on horse and jockey performance, which provides insight into how future work into this field could help influence practice to reduce injury.

Abstract

Generally clear and summarises study well.

Line 22: please remove hyphen for upper body

Line 38: suggest amending to significance was set at p<0.05

Line 39: please amend to indicated

Please be consistent with use of < or ≤ throughout

Line 48-49: it would be beneficial to interpret to some extent what the amount of post-hoc significant differences translate to in practice / interpretation for the reader

Line 50: Please amend to The results support…

Introduction

Interesting and informative introduction to the study, establishes clear rationale and provides background to support readers less familiar with this field.

Materials and methods

Comprehensive and clear

Line 213-216: please amend to make past tense

Line 244: please amend to make past tense

Results

There is a lot of information presented here and while I can understand the decision to present in figures / tables then summarise, I would suggest the initial sections (prior to summary / line 344) would benefit from some brief interpretative text to help the reader interpret the data presented more and lead into the ‘in summary’ ending.

Line 284: please include standard deviation for mean as well to substantiate low variance in SF

Line 308-311 and 319-324: please amend to make past tense as these data / inferences relate to your data

Line 360-362: some narrative explanation of results evaluating the post hoc differences to show why #15% more based on rider data compared to horse data would be beneficial

Tables/ figures: please include legends and ensure all abbreviations used within tables and figures are provided within these (including those in supplementary files)

Discussion

Interesting and relevant discussion; could be worth considering rider experience in here as well as assuming used BRS students so could argue who may not possess as much postural stability as experienced jockeys

Line 394: upper body doesn’t needs to be hyphenated

Conclusion summarises key take home messages

Reviewer #2: This article presents information on how horses hooves interact with different ground surfaces and the effect of this on horse and rider displacement. It presents valuable information on the different COM displacements according to different shoe and surface conditions.

The authors claims are properly presented within the context of the previous literature. A sentence linking how the COM displacements of horse and rider may relate to the ground reaction forces experienced by the horses limb (and thus injury) would clarify the introduction further. Greater attention to grammar used in some sections would ease reading of the manuscript. The terms DV/CC/ML are interspersed with their colloquial alternatives and sometimes x,y or z axis, it would be easier to read if one or other of these terms were used consistently throughout. The titles of some references need to be in sentence case, not capitalised.

Abstract

Lines 48-49 – Unclear what is meant by the number of significant differences from the post hoc – could you rewrite to explain what is meant by this?

Line 50 – affect not effect

Lines 50-59 – is there a combination of shoe/surface that results in a more stable dyad? What have you defined as more stable/less likely to induce injury?.

Introduction

Line 80 – is inertia really decreased?. Though the jockey is moving little wrt world inertial frame, his joints are moving a lot within the bounds of the dyad to attenuate the forces produced by the horses movement, so is not actually holding a static posture.

Methods

It is a shame that the accelerometers on horse and rider could not be synchronised to find movement differences, but the information gained from what was available was useful. How was gravity accounted for in the accelerometers ?– as when the horse (and particularly the jockey) moves, gravity would act in part on different axes – was it assumed that the jockey’s back remained horizontal to the ground during the entire trial?. The double integration would compound these errors, and there should be a mention of this in the discussion.

Results

Line 296 – Does the approx. of 40% need SD/SE?

Lines 358 – a bit more explanation of this metric may be required, does this imply that the rider experiences more displacement differences than the horse?

Figures 9-11. Some of the absolute displacements look like they’re the same values, but the subscript indicates they have significantly different EMMs, how can this be?

Discussion

413-417 Are you stating the same thing twice?

Line 423 – consistency in use of axes terms (z and x correspond to which axes?)

Line 429 – Be clear that the hooves sink more into the artificial surface on impact rather than the turf. Is there any link to greater displacements with horse stability/injury? Are you equating a larger DV displacement with energy loss?

Line 488 - Does the CC displacement equate to a greater propulsive force? This seems to be indicated, and indicates that artificial surface may result in less wasted movement (DV) and thus faster racing?

Would be nice to have a bit of discussion on how the difference in displacements might affect the horse and riders stability or traction, linking back to your reasons for doing the study.

6. PLOS authors have the option to publish the peer review history of their article (what does this mean?). If published, this will include your full peer review and any attached files.

Reviewer #1: No

Reviewer #2: No

---

## [Author Response · Author response to Decision Letter 0]

19 Aug 2021

Response to Reviewers 

Journal Requirements:

We have checked our manuscript against the PLOS ONE style requirements.

'I have read the journal's policy and the authors of this manuscript have the following competing interests: Thilo Pfau is owner of EquiGait Ltd providing equine gait analysis products and services.'

We note that one or more of the authors are employed by a commercial company: EquiGait Ltd and James Coburn AWCF Ltd.

Yes, the funding organization (HBLB) did not play a role in the study design, data collection and analysis, decision to publish, or preparation of the manuscript and only provided financial support in the form of authors' salaries and/or research materials.

The following has been added to our manuscript. “The funder provided support in the form of salaries for authors [KH (full-time) and KK, JC, PD, HC, DH, LB and LH (during data collection only)], but did not have any additional role in the study design, data collection and analysis, decision to publish, or preparation of the manuscript. The specific roles of these authors are articulated in the ‘author contributions’ section.

It did not play a role.

We have added the following sentence to the Competing Interests Statement: “JC is the owner of James Coburn AWCF Ltd, which employs JC, HC, DH and LB. This does not alter our adherence to PLOS ONE policies on sharing data and materials.”

Our cover letter has been updated accordingly. 

The Horan et al. (in press) reference in the original manuscript has been updated to Horan et al. (2021). 

The revised manuscript also includes the following additional references:

Introduction:

Ruina A, Bertram JEA, Srinivasan M. A collisional model of the energetic cost of support work qualitatively explains leg sequencing in walking and galloping, pseudo-elastic leg behavior in running and the walk-to-run transition. J Theor Biol. 2005;237: 170–192. doi:10.1016/j.jtbi.2005.04.004

Bertram JEA, Gutmann A. Motions of the running horse and cheetah revisited: fundamental mechanics of the transverse and rotary gallop. J R Soc Interface. 2009;6: 549–559. doi:10.1098/rsif.2008.0328

Pfau T, Witte TH, Wilson AM. Centre of mass movement and mechanical energy fluctuation during gallop locomotion in the Thoroughbred racehorse. J Exp Biol. 2006;209: 3742–3757. doi:10.1242/jeb.02439

Discussion:

Gonzalez, M.E. and Sarabon, N., 2020. Muscle modes of the equestrian rider at walk, rising trot and canter. PLoS ONE 15: e0237727. https://doi.org/10.1371/journal.pone.0237727

Hitchens, P.L., Blizzard, C.L., Jones, G., Day, L.M. and Fell, J., 2012. The association between jockey experience and race-day falls in flat racing in Australia. Injury Prevention 18: 385-391. https://doi. org/10.1136/injuryprev-2011-040255.

Horan, K., Coburn, J., Kourdache, K., Day, P., Harborne, D., Brinkley, L., Carnall, H., Hammond, L., Peterson, M., Millard, S. and Thilo Pfau. 2021, in review. Influence of speed, ground surface and shoeing condition on hoof breakover duration in galloping Thoroughbred racehorses. https://doi:10.20944/preprints202108.0323.v1

Legg, K.A., Cochrane, D.J., Bolwell, C.F., Gee, E.K. and Rogers, C.W., 2020. Incidence and risk factors for race-day jockey falls over fourteen years. Journal of Science and Medicine in Sport 23: 1154- 1160. https://doi.org/10.1016/j.jsams.2020.05.015

Parkin, T.D.H., Clegg, P.D., French, N.P., Proudman, C.J., Riggs, C.M., Singer, E.R., Webbon, P.M. and Morgan, K.L., 2004. Race- and course-level risk factors for fatal distal limb fracture in racing Thoroughbreds. Equine Veterinary Journal 36: 521-526.

Setterbo JJ, Fyhrie PB, Hubbard M, Upadhyaya SK, Stover SM. Dynamic properties of a dirt and a synthetic equine racetrack surface measured by a track-testing device. Equine Vet J. 2013;45: 25–30. doi:10.1111/j.2042-3306.2012.00582.x

Terada, K., 2000. Comparison of head movement and EMG activity of muscles between advanced and novice horseback riders at different gaits. Journal of Equine Science 11: 83-90. https://doi.org/10.1294/ jes.11.83

3. We note that Figure 2 includes an image of a [patient / participant / in the study]. 

The image is of author KK who has given written informed consent to publish these case details. We have added the requested statement to our methods (ethics) section. KK has also signed a consent form.

Data were submitted along with the manuscript in an Excel File. We have added this information to our cover letter.

Additional Editor Comments (if provided):

Thank you for your submission. Both reviewers have identified a number of minor editorial revision are required before acceptance.

Reviewers' comments:

Reviewer's Responses to Questions

Comments to the Author

1. Is the manuscript technically sound, and do the data support the conclusions?

Reviewer #1: Yes

Reviewer #2: Yes

2. Has the statistical analysis been performed appropriately and rigorously?

Reviewer #1: Yes

Reviewer #2: Yes

3. Have the authors made all data underlying the findings in their manuscript fully available?

Reviewer #1: Yes

Reviewer #2: Yes

4. Is the manuscript presented in an intelligible fashion and written in standard English?

Reviewer #1: Yes

Reviewer #2: Yes

5. Review Comments to the Author

Reviewer #1: A well-presented and really interesting applied study that provides some preliminary insight into the impact of surfaces and shoe types on horse and jockey performance, which provides insight into how future work into this field could help influence practice to reduce injury.

We would like to thank reviewer 1 for their comments and suggestions to improve our manuscript. 

Abstract

Generally clear and summarises study well.

Line 22: please remove hyphen for upper body

This correction has been made.

Line 38: suggest amending to significance was set at p<0.05

This correction has been made.

Line 39: please amend to indicated

This correction has been made.

Please be consistent with use of < or ≤ throughout

Our use of < and ≤ is correct here. The difference reflects the fact that a SPSS output of “0.000” means a value less than 0.0005, whereas a description of a group of parameters for which one or more p values are greater than 0.0005 must have p values equal to or less than the stated value. For example, p≤0.008 means none of the p values summarised exceeded 0.008. We hope this clarifies our approach.

Line 48-49: it would be beneficial to interpret to some extent what the amount of post-hoc significant differences translate to in practice / interpretation for the reader

On reflection, we have decided to omit this detail about the post-hoc tests in the abstract. We are limited on words and feel this information is not critical here. As the reviewer points out, without further explanation as to how the numbers were acquired (for example, only the estimated marginal means of significant shoe-surface interactions were investigated, as detailed in the results, giving rise to a differing denominator for horse and rider) this sentence is difficult to interpret. The implications of the differing number of significant results in the post-hoc tests of horse and rider are discussed in detail in the discussion. 

Line 50: Please amend to The results support…

This correction has been made.

Introduction

Interesting and informative introduction to the study, establishes clear rationale and provides background to support readers less familiar with this field.

Materials and methods

Comprehensive and clear

We thank the reviewer for their positive feedback on our introduction and methods.

Line 213-216: please amend to make past tense

This has been amended.

Line 244: please amend to make past tense

This has been amended.

Results

There is a lot of information presented here and while I can understand the decision to present in figures / tables then summarise, I would suggest the initial sections (prior to summary / line 344) would benefit from some brief interpretative text to help the reader interpret the data presented more and lead into the ‘in summary’ ending.

We have now added the following sentences at the beginning of the ‘Displacement minima and maxima’ section. 

“Here, we first present the collective data of all shoe-surface conditions to assess the magnitude of the tri-axial displacements and look for general patterns in horse and rider movements. We then explore the data in more detail to see whether shoe or surface had any additional impact on the results.”

We have also restructured the ‘General trends’ sub-section so that the mean values (Table 2) come first. We have removed the words “In summary” from line 364, because this was not appropriate here. 

Line 284: please include standard deviation for mean as well to substantiate low variance in SF

The stride frequency for the horse and rider data sets are now both reported with SD (RM line 296).

Line 308-311 and 319-324: please amend to make past tense as these data / inferences relate to your data

This has been amended in each case.

Line 360-362: some narrative explanation of results evaluating the post hoc differences to show why #15% more based on rider data compared to horse data would be beneficial

We apologise that this was unclear in our original manuscript. There are eight possible shoe-surface combinations arising from two surfaces and four shoe types. When these are sequentially compared against one another this gives rise to 28 comparisons per axis direction. As there are six possible directions reflecting movement in both positive and negative directions for CC, ML and DV displacement, there are 168 comparisons possible. For the rider, shoe-surface interactions only significantly affected displacement in five of these six directions, so only 140 comparisons were made. Of these comparisons, the EMMs were significantly different in 83 cases. For the horse, three directions were significantly affected, so 84 comparisons were made between shoe-surface conditions. In this case, 55 were found to be significantly different. We have now added the following to our revised manuscript (lines 380-391).

“Post-hoc tests were performed when the interaction was significant. As there were eight possible shoe-surface combinations arising from two surfaces and four shoe types, sequentially comparing these against one another gave rise to 28 comparisons per axis direction. With six possible directions reflecting movement in both positive and negative directions for CC, ML and DV displacement, there was a maximum of 168 comparisons possible. However, for the rider, for whom shoe-surface interactions only significantly affected displacement in five of the six directions, 140 comparisons were made. Of these comparisons, the EMMs were significantly different in 83 cases. For the horse, 84 comparisons were made for the three directions that were significantly affected and 55 shoe-surface EMMs were found to be significantly different (Supporting Information, Table S3).”

In addition, we have now included the post-hoc results within the supplementary information for both shoe-type (Table S2) and shoe-surface combinations (Table S3). This also helps to clarify the comparisons made. 

Tables/ figures: please include legends and ensure all abbreviations used within tables and figures are provided within these (including those in supplementary files)

The following amendments have been made to the figures.

Figure 1: Letters A-F have been added on to images. The abbreviation ‘BRS’ has been changed to ‘British Racing School’ in the caption. The order of the shoes has been switched for consistency with data tables (i.e. alphabetical).

Figure 2: The caption has been amended so the terms x, y and z are replaced with CC, ML and DV.

Figure 3: Text has been enlarged on some of the subplots. The text has been amended so the terms x and z are replaced with CC and DV. The abbreviation COM in the caption has been changed to centre of mass. CC and DV are explained in the caption.

Figure 4: We have deleted reference to Supplementary Information in the caption as we removed the discussion from here on why speed and acceleration data could not be aligned (removed prior to original manuscript submission).

Figure 5: No changes

Figure 6: The terms DV and CC are expanded in the caption.

Figure 7: Dot at the origin has been deleted. The 0 on the x-axis has been changed to 0.0. The abbreviations CC and DV are expanded in the caption.

Figure 8: No changes

Figure 9: No changes

Figure 10: The abbreviations CC, ML and DV are explained in the caption.

Figure 11: The abbreviations CC, ML and DV are explained in the caption. The following sentence was also added to the caption: “Note that pairwise comparisons between shoe-surface combinations in a particular axis direction were only made when there was found to be a significant interaction between shoe and surface.” This is to also help clarify our approach, in response to the query raised by the reviewer above.

Figure 12: The abbreviations EMM, CC and DV are expanded in the caption. The lines of best-fit are now dashed, for consistency with Figure 13.

Figure 13: The abbreviations EMM, CC and DV are expanded in the caption.

Figure S1: No changes

Figure S2: Axes on figures updated so x, y, and z are replaced with CC, ML and DV. These terms are explained in the caption.

Figure S3: Axes on figures updated so x, y, and z are replaced with CC, ML and DV. These terms are explained in the caption.

The following amendments have been made to the tables.

Table 1: Word “gallop” added to table caption. Asterisks deleted from column 4. Order of data restructured to match order in stats results tables.

Table 2: CC, ML and DV explained in caption.

Table 3: CC, ML and DV explained in caption.

Table 4: CC, ML and DV explained in caption.

Table 5: CC, ML and DV explained in caption.

Table 6: CC, ML and DV explained in caption.

Table S1: Order of columns changed for consistency with stats results tables e.g. Table S2 and S3.

Table S2: New to the revised manuscript. Shows post-hoc pairwise comparisons (with Bonferroni correction) for shoe type. Significant differences are highlighted in bold. 

Table S3. New to the revised manuscript. Post-hoc pairwise comparisons (with Bonferroni correction) for shoe-surface combinations. Significant differences are highlighted in bold. 

Discussion

Interesting and relevant discussion; could be worth considering rider experience in here as well as assuming used BRS students so could argue who may not possess as much postural stability as experienced jockeys

The riders were not students but did only have a limited number of professional years of experience, as detailed in the methodology.

Our discussion had already considered the issue of rider skill in the following places. To this, we have added the text underlined.

Lines 454-457 (revised manuscript): “As the same horse-rider combinations were assessed on the different surfaces and with different shoes we created a ‘paired’ comparison, which minimized the risk of individual horse (and jockey) related characteristics confounding results, such as skill or age.”

Lines 512-524 (revised manuscript) “Riding experience and skill may influence rider ability to follow such subtleties in the movement patterns of their horse [93,94 Terada et al., 2000], or to affect changes in gait by controlling their mass distribution, as has been reported in lameness examinations [95]. Jockeys with more experience are reported to have a lower risk of falling, and their horses have a lower risk of fatal limb fractures compared to jockeys with less experience (Hitchens et al., 2012; Legg et al., 2020; Parkin et al., 2004). This may be because advanced riders anticipate horse movement at a neuromuscular level with more defined and relaxed contralateral muscular activation patterns than novice riders (Gonzalez and Sarabon, 2020; Terada et al., 2000). The riders in this study only had a limited number of professional years of experience, so their movement patterns may not be directly transferable to those of a more experienced jockey. Rider pPain or stiffness may also limit rider their ability to follow horse movements [5,6] and the natural biomechanical asymmetries of the riders used here was not assessed. However, since the horse-rider pairs remained fixed it is not expected that these factors will have skewed our interpretation of horse and rider response to the difference shoe-surface combinations, although it is possible that the nuances of the response may differ amongst riders.results.”

Hitchens PL, Leigh Blizzard C, Jones G, Day LM, Fell J. The association between jockey experience and race-day falls in flat racing in Australia. Inj Prev. 2012;18: 385–391. doi:10.1136/injuryprev-2011-040255

Legg KA, Cochrane DJ, Bolwell CF, Gee EK, Rogers CW. Incidence and risk factors for race-day jockey falls over fourteen years. J Sci Med Sport. 2020;23: 1154–1160. doi:10.1016/j.jsams.2020.05.015

Parkin TDH, Clegg PD, French NP, Proudman CJ, Riggs CM, Singer ER, et al. Race- and course-level risk factors for fatal distal limb fracture in racing Thoroughbreds. Equine Vet J. 2004;36: 521–526. 

González E.M, Šarabon N. Muscle modes of the equestrian rider at walk, rising trot and canter. PLoS One. 2020;15: e0237727. doi:10.1371/JOURNAL.PONE.0237727

Terada K. Comparison of head movement and EMG activity of muscles between advanced and novice horseback riders at different gaits. J Equine Sci. 2000;11: 83–90. doi:10.1294/JES.11.83

Line 394: upper body doesn’t needs to be hyphenated

This correction has been made.

Conclusion summarises key take home messages

Reviewer #2: This article presents information on how horses hooves interact with different ground surfaces and the effect of this on horse and rider displacement. It presents valuable information on the different COM displacements according to different shoe and surface conditions.

The authors claims are properly presented within the context of the previous literature. A sentence linking how the COM displacements of horse and rider may relate to the ground reaction forces experienced by the horses limb (and thus injury) would clarify the introduction further. Greater attention to grammar used in some sections would ease reading of the manuscript. The terms DV/CC/ML are interspersed with their colloquial alternatives and sometimes x,y or z axis, it would be easier to read if one or other of these terms were used consistently throughout. The titles of some references need to be in sentence case, not capitalised.

We would like to thank reviewer 2 for their comments and suggestions to improve our manuscript. 

We now include the following additional information in our introduction to suggest how the COM displacements of horse and rider may relate to the ground reaction forces experienced by the horses’ limbs, and the associated risk of injury. (Key changes are underlined)

(Revised manuscript lines 78-94) “The horses’ limbs act in sequence to redirect their COM [7]. The first footfalls (hindlimbs) accelerate the horse, propelling the COM forwards, and the later ones (forelimbs) decelerate the horse and apply vertical impulse to the COM [8]. Energy is lost during the stance phase of the limbs and is a function of the change in the angle of the trajectory of the COM [7]. The leading forelimb is thought to be the most important for redirecting the COM, as a result of cranio-caudal (CC) deceleration, vertical acceleration and an increase in potential energy of the COM occurring during the stance phase of this limb [9]. On a stride per stride basis, jockey kinematics adjust to accommodate the changes in translational and rotational upper-body movements of the horse and thereby maintain stability [10]. For example, during stance of the leading hindlimb at gallop, the horse’s trunk and jockey’s pelvis both displace laterally and roll away from the side of this leg [10]. Force data from stirrups indicate that jockeys push away from the stirrup on the non-lead side at this time to maintain the position of their COM close to the horse’s midline and balance themselves [10]. A slight delay in the dorso-ventral (DV) displacement of a jockey’s pelvis occurs relative to the horse, but medio-lateral (ML) movements are in phase [10]. Importantly, as the jockey’s COM moves out of phase with the horse in the CC-axis [10,11], this is when they are most unstable [10]. It is not known at which point in the stride cycle jockeys are most susceptible to injury, but it is plausible that it would be when displacements peak in the CC-axis.”

The additional references in this paragraph are as follows.

Ruina A, Bertram JEA, Srinivasan M. A collisional model of the energetic cost of support work qualitatively explains leg sequencing in walking and galloping, pseudo-elastic leg behavior in running and the walk-to-run transition. J Theor Biol. 2005;237: 170–192. doi:10.1016/j.jtbi.2005.04.004

Bertram JEA, Gutmann A. Motions of the running horse and cheetah revisited: fundamental mechanics of the transverse and rotary gallop. J R Soc Interface. 2009;6: 549–559. doi:10.1098/rsif.2008.0328

Pfau T, Witte TH, Wilson AM. Centre of mass movement and mechanical energy fluctuation during gallop locomotion in the Thoroughbred racehorse. J Exp Biol. 2006;209: 3742–3757. doi:10.1242/jeb.02439

We have revised some sections of text to past tense to improve the flow of manuscript, also in response to Reviewer 1.

We have altered our manuscript so only the terms DV/CC/ML are included (without the confusion of x,y,z).

We have checked the references and amended accordingly. Thank you for bringing this to our attention.

We provide an additional point by point response to the other suggestions of Reviewer 2 below.

Abstract

Lines 48-49 – Unclear what is meant by the number of significant differences from the post hoc – could you rewrite to explain what is meant by this?

We apologise that this was not clear. The revised manuscript no longer contains this information (see also response to reviewer 1). We have provided additional detail on how the number of significant differences and comparisons arose in the results section of the revised manuscript. 

Line 50 – affect not effect

This correction has been made.

Lines 50-59 – is there a combination of shoe/surface that results in a more stable dyad? What have you defined as more stable/less likely to induce injury?

We have added the following to our revised manuscript. 

Abstract, RM lines 53-55: 

“The artificial surface and steel shoes provoked the least cranio-caudal axis movement of the jockey, so may promote greatest stability.”

Discussion, RM lines 473-477: It is important to consider that greater CC movement of the rider could compromise stability, because their COM moves out of phase with the horse in this axis, in contrast to the DV and ML axes where the horse and rider movements are almost in phase [10,11]. As such, jockey stability could be considered to be greater on the artificial surface, where CC movements were lower compared to turf under the ground conditions studied.

& RM lines 488-489: “This reduction in CC displacement may also indicate improved rider stability under the steel shoeing condition.”

Introduction

Line 80 – is inertia really decreased?. Though the jockey is moving little wrt world inertial frame, his joints are moving a lot within the bounds of the dyad to attenuate the forces produced by the horses movement, so is not actually holding a static posture.

We are not saying that the position is good for the jockey, on the contrary it is ‘costly’ for the jockey. 

Methods

It is a shame that the accelerometers on horse and rider could not be synchronised to find movement differences, but the information gained from what was available was useful. How was gravity accounted for in the accelerometers ?– as when the horse (and particularly the jockey) moves, gravity would act in part on different axes – was it assumed that the jockey’s back remained horizontal to the ground during the entire trial?. The double integration would compound these errors, and there should be a mention of this in the discussion.

The horse and jockey were only assessed at the gallop, where the jockey always adopted the ‘martini glass’ posture. The general orientation of the axes therefore remained fixed throughout trials, as per Figure 2. It is true that the precise orientation of the axes (and hence effect of gravity) will have differed to some extent throughout a stride cycle. However, at the respective minima and maxima displacements along each axis, which are the measurements of interest here, this orientation will be consistent. 

Results

Line 296 – Does the approx. of 40% need SD/SE?

The value of 40% had reflected a consideration of all axes together, and the visual depiction of an extract of data in Figure 4. However, to improve clarity we have deleted this sentence and replaced with the following.

“Rider displacements were reduced relative to the horse in all directions. This difference was proportionally greatest in the cranio-caudal axis, particularly in the forwards direction (Table 2, Fig 4).”

The SD/SE are provided for the mean values reported in Table 2 for each axis. 

Lines 358 – a bit more explanation of this metric may be required, does this imply that the rider experiences more displacement differences than the horse?

We assume the reviewer is referring here to the post-hoc test results (lines 361-363 in the original manuscript)? We now provide the following explanation as to how this was quantified.

“Post-hoc tests were performed when the interaction was significant. As there were eight possible shoe-surface combinations arising from two surfaces and four shoe types, sequentially comparing these against one another gave rise to 28 comparisons per axis direction. With six possible directions reflecting movement in both positive and negative directions for CC, ML and DV displacement, there was a maximum of 168 comparisons possible. However, for the rider, for whom shoe-surface interactions only significantly affected displacement in five of the six directions, 140 comparisons were made. Of these comparisons, the EMMs were significantly different in 83 cases. For the horse, 84 comparisons were made for the three directions that were significantly affected and 55 shoe-surface EMMs were found to be significantly different (Supporting Information, Table S3).”

Figures 9-11. Some of the absolute displacements look like they’re the same values, but the subscript indicates they have significantly different EMMs, how can this be?

As detailed in the results Tables 3-5, the absolute differences in mean displacements between conditions are often small, particularly in the ML axis. However, these differences were still found to be significant in some cases. Please note that in the case of Figure 11, only the significant interactions (marked *) were investigated in the post-hoc tests (as detailed above). For clarity this information has also been added to the Figure 11 caption: “Note that pairwise comparisons between shoe-surface combinations in a particular axis direction were only made when there was found to be a significant interaction between shoe and surface.”

Discussion

413-417 Are you stating the same thing twice?

Thank you for bringing this to our attention. We have deleted the second sentence.

Line 423 – consistency in use of axes terms (z and x correspond to which axes?)

We have altered our manuscript so only the terms DV/CC/ML are included (without the confusion of x, y and z). Figures have also been updated.

Line 429 – Be clear that the hooves sink more into the artificial surface on impact rather than the turf. Is there any link to greater displacements with horse stability/injury? Are you equating a larger DV displacement with energy loss?

We have added “on the artificial surface” to RM line 459 to improve clarity. 

Greater cranio-caudal movement for the jockey may be linked to reduced stability because they are moving out of phase with the horse in this axis.

We have updated this paragraph to the following.

RM lines 458-481 “Here, significantly greater vertical displacements were apparent for horses galloping on the artificial compared to the turf surface (Table 3, Fig 9). This may reflect the hooves sinking more at impact on the artificial surface, combined with greater propulsive forces being generated during the latter part of stance. Artificial surfaces may undergo a high degree of elastic deformation [85] and therefore return more energy to the hoof. A more rapid hoof breakover on the artificial surface [86] would be consistent with the notion that push-off forces, which would impose vertical COM movement, are stronger on this track, relative to turf. However, significantly higher CC horse displacement occurred on turf (Table 3, Fig. 9). This could be linked to increased grip at the hoof-surface interface, as this would increase CC forces [56], and may suggest that surface traction has the dominant control on CC movement rather than any horizontal propulsive forces generated during breakover. Nevertheless, further work is needed to establish which surface conferred the greatest grip, as reduced breakover times on the artificial track could suggest the hooves had greater purchase on this surface (Horan et al., 2021, in review). It is also plausible that the softer nature of the artificial track caused the toe to rotate into this surface more easily than on turf [86]; this effect may help explain both why hoof breakover took longer on the turf track and why the associated upper body movements were more pronounced in the CC direction on turf. The patterns in horse displacements were mirrored in rider displacements. It is important to consider that greater CC movement of the rider could compromise stability. This is because their COM moves out of phase with the horse in this axis, in contrast to the DV and ML axes where the horse and rider movements are almost in phase [8,12]. As such, jockey stability could be considered to be greater on the artificial surface, where CC movements were lower compared to turf, under the ground conditions studied. Stability is likely to be associated with safety, and jockey perception of safety and grip are strongly correlated [64]. It is therefore interesting to note that jockeys perceived grip to be enhanced on the artificial surface [64], which may suggest they attribute the experience of lowered CC movement to increased grip at the hoof-surface interface. 

New references:

Horan, K., Coburn, J., Kourdache, K., Day, P., Harborne, D., Brinkley, L., Carnall, H., Hammond, L., Peterson, M., Millard, S. and Thilo Pfau (2021, in review) Influence of speed , ground surface and shoeing condition on hoof breakover duration in galloping Thoroughbred racehorses. https://doi:10.20944/preprints202108.0323.v1

Setterbo JJ, Fyhrie PB, Hubbard M, Upadhyaya SK, Stover SM. Dynamic properties of a dirt and a synthetic equine racetrack surface measured by a track-testing device. Equine Vet J. 2013;45: 25–30. doi:10.1111/j.2042-3306.2012.00582.x

Line 488 - Does the CC displacement equate to a greater propulsive force? This seems to be indicated, and indicates that artificial surface may result in less wasted movement (DV) and thus faster racing?

Would be nice to have a bit of discussion on how the difference in displacements might affect the horse and riders stability or traction, linking back to your reasons for doing the study.

We now have hoof breakover data from this sample of horses, which indicates a faster breakover occurs on the artificial track. The artificial track actually shows increased DV movement, which may reflect the generation of stronger propulsive forces (acting both forwards and upwards).

However, since CC movements were greater on turf this could indicate that the dominant control on CC movement occurs on impact, and is governed by the amount of traction offered. 

The paragraph was revised, as detailed above (RM lines 458-481).

6. PLOS authors have the option to publish the peer review history of their article (what does this mean?). If published, this will include your full peer review and any attached files.

Do you want your identity to be public for this peer review? For information about this choice, including consent withdrawal, please see our Privacy Policy.

Reviewer #1: No

Reviewer #2: No

---

## [Editor Report · Decision Letter 1]

13 Sep 2021

The effect of horseshoes and surfaces on horse and rider centre of mass displacements at gallop

PONE-D-21-05125R1

Dear Dr. Horan,

We’re pleased to inform you that your manuscript has been judged scientifically suitable for publication and will be formally accepted for publication once it meets all outstanding technical requirements.

Kind regards,

Chris Rogers

Academic Editor

PLOS ONE

Additional Editor Comments (optional):

Thank you for your edits to the manuscript. It may now proceed with the publication process.
---

## [Editor Report · Acceptance letter]

15 Nov 2021

PONE-D-21-05125R1 

The effect of horseshoes and surfaces on horse and jockey centre of mass displacements at gallop 

Dear Dr. Horan:

I'm pleased to inform you that your manuscript has been deemed suitable for publication in PLOS ONE. Congratulations! Your manuscript is now with our production department. 

Kind regards, 

on behalf of

Dr. Chris Rogers 

Academic Editor

PLOS ONE